# scMINER: a mutual information-based framework for clustering and hidden driver inference from single-cell transcriptomics data

Qingfei Pan[1,9], Liang Ding[1,9], Siarhei Hladyshau[1,9], Xiangyu Yao[1,9], Jiayu Zhou[1], Lei Yan[1], Yogesh Dhungana[1,2], Hao Shi[3], Chenxi Qian[1], Xinran Dong[4], Chad Burdyshaw[5], Joao Pedro Veloso[1], Alireza Khatamian[1], Zhen Xie[1,6], Isabel Risch[1,3], Xu Yang[1], Jiyuan Yang[1], Xin Huang[1,7], Jason Fang[1], Anuj Jain[1], Arihant Jain[1], Michael Rusch[1], Michael Brewer[5], Junmin Peng[8], Koon-Kiu Yan[1], Hongbo Chi[2,3] & Jiyang Yu[1,2]✉

Single-cell transcriptomics data present challenges due to their inherent stochasticity and sparsity, complicating both cell clustering and cell type-specific network inference. To address these challenges, we introduce **scMINER** (single-cell Mutual Information-based Network Engineering Ranger), an integrative framework for unsupervised cell clustering, transcription factor and signaling protein network inference, and identification of hidden drivers from single-cell transcriptomic data. scMINER demonstrates superior accuracy in cell clustering, outperforming five state-of-the-art algorithms and excelling in distinguishing closely related cell populations. For network inference, scMINER outperforms three established methods, as validated by ATAC-seq and CROP-seq. In particular, it surpasses SCENIC in revealing key transcription factor drivers involved in T cell exhaustion and Treg tissue specification. Moreover, scMINER enables the inference of signaling protein networks and drivers with high accuracy, which presents an advantage in multimodal single cell data analysis. In addition, we establish scMINER Portal, an interactive visualization tool to facilitate exploration of scMINER results.

Cell identity is determined and regulated by the wiring and rewiring of networks of transcription factors (TFs) and signaling proteins (SIGs)[1]. Identifying the "drivers" of these networks is crucial for understanding cellular plasticity and dynamics as well as for determining therapeutic targets for diseases[2]. The prediction of downstream target genes, also known as gene network inference, is currently a major obstacle in identifying these drivers. Single-cell RNA sequencing (scRNA-seq) technology has made it possible to profile cell types and states at an unprecedented resolution, providing information about cellular

heterogeneity unattainable from bulk transcriptomic data, in which the signal of individual cells in a sample is lost. Over a dozen algorithms for inferring gene regulatory networks from scRNA-seq data have been developed[3–16]. However, the overall performance of these approaches, especially the accuracy, is less than ideal[17]. Some major challenges persist in cell-type-specific network rewiring and driver identification.

The first challenge is to accurately group cell populations to achieve high purity and homogeneity. Most of the existing single-cell

---

clustering algorithms[18] rely on linear dimensionality reduction methods; as a result, these methods may not capture non-linear cell-cell dependencies. For example, one of the most popular tools, Seurat[19], reduces the dimensionality of the data using principal component analysis (PCA), which is mathematically an orthogonal linear transformation[20]. Another shortcoming common among existing clustering tools is their selective use of highly variable genes, rather than all genes, for clustering analysis. While this strategy can improve the processing speed, the selection of top variable features is biologically arbitrary and may result in further information loss. These methods are limited in their ability to detect rare clusters or distinguish between similar cell states[18,21], complicating network inference.

The second challenge is accurately estimating gene-gene dependency from scRNA-seq data, which is inherently stochastic and sparse[22,23]. Most existing methods of network inference[6,7,9,10,13–15,24,25] measure the similarity of genes using linear regression metrics (e.g., Pearson correlation coefficient) that assume linearity, homoscedasticity, independence and normality-features not common of single-cell transcriptomic data[22]. Some methods for network inference[12,26–31] integrate scRNA-seq data with data from other modalities to improve the accuracy of their predictions. For example, the commonly used method SCENIC[12] uses TF binding motif databases and co-expression analysis to reconstruct TF-target networks and infer TF activity for subsequent clustering analysis. LINGER[32], a recently developed method that incorporates single-cell multiome data with atlas-scale bulk data, has also achieved better accuracy in network inference. The identification of TF master regulators has a long history, and quite a few methods like co-expression and matrix factorization have been proposed[26,33,34]. These methods use the genome-wide expression data and infer the activity of a transcription factor by the expression of downstream target genes. For example, SCENIC scores the activity of a TF by calculating the enrichment of its target genes ranked by gene expression values. Nevertheless, the accuracy of GRN (gene regulatory network) inference has remained low, slightly higher than random predictors[17].

Finally, while tools have been developed for inferring TF regulatory networks and drivers from scRNA-seq data, there are no comparable methods for signaling networks and drivers. This is due, in part, to the fact that many signaling drivers are not differentially expressed at the mRNA or protein level; rather, they are altered by post-translational modifications (PTMs, e.g., kinase) and other mechanisms[35], leading them to be referred to as "hidden drivers"[36]. Because signaling networks are not easily inferred from gene expression profiles, few tools have been developed to infer signaling networks from bulk samples[37–39], and none to infer cell-type-specific signaling networks from single-cell transcriptomics data. Signaling networks have been recognized to impart emergent properties associated with differentiation and cell state transition[40,41]. In addition, the druggable nature of signaling proteins makes them an attractive perturbation target for therapeutic interventions[42,43]. The availability of tools for inferring cell-type-specific signaling networks from single-cell data is therefore a significant unmet need.

To address these challenges, we developed single-cell Mutual Information-based Network Engineering Ranger (scMINER) an integrative framework for unsupervised cell clustering, TF and SIG network inference, and hidden driver identification from single-cell transcriptomics data.

## Results

### The scMINER framework

scMINER provides a user-friendly interface for the comprehensive analysis of scRNA-seq data, including data preprocessing, Mutual Information-based Clustering Analysis (MICA), cluster-specific network inference, cell type-specific hidden driver identification, and data visualization and sharing (Fig. 1). scMINER takes a single-cell gene expression matrix as its input, performing initial quality control and data filtration steps before initiating MICA. MICA leverages mutual information (MI) to measure the distance between cells, enabling the characterization of nonlinear dependence in gene expression distributions among cells. To maximize the accuracy of clustering while maintaining flexibility to handle large datasets, MICA selectively employs one of two commonly used scRNA-seq data clustering strategies depending on the size of the dataset[18,23]. For smaller datasets (default <5,000 cells), MICA applies consensus k-means-based clustering similar to SC3s[44], which iteratively identifies the globally optimal k clusters and uses a consensus approach to enhance robustness; this strategy is slower, but more accurate. For larger datasets, MICA employs a faster, graph-based approach as used by Seurat[19] and Scanpy[45] that constructs a heuristic cell-cell graph before performing community detection. To reduce the number of separate dimensions in the data, MICA incorporates nonlinear multidimensional scaling (MDS) in k-means-based clustering and graph embedding (GE) in graph-based approach.

Using the clustering results generated by MICA, scMINER can reverse-engineer intracellular gene networks for each cluster using a modified version of SJARACNe[46], an MI-based algorithm that was originally developed for bulk omics data but has been re-parameterized to handle single-cell transcriptomics data. scMINER also provides a meta-cell-based approach[47] to diminish the effect of the sparsity of scRNA-seq data by aggregating the gene expression profiles of similar cells. Using the data-driven and cell type-specific networks, scMINER can transform single-cell gene expression matrices into single-cell activity profiles that can be used to identify cluster-specific TFs and SIGs, including hidden drivers that show changes at the activity but not expression level.

In addition to the scMINER framework, we have established the scMINER Portal (https://scminer.stjude.org), a web-based, centralized data platform for visualizing, exploring and sharing single-cell transcriptomics studies (Supplementary Fig. 1). The portal integrates several products, including the Neo4j graph database management system[48], a robust search engine, and cutting-edge data visualization libraries[49–53], that allow users to explore cell clusters, investigate marker genes at the expression and activity levels, visualize gene networks, and perform intra- and cross-study keyword searches.

### scMINER outperforms popular single-cell clustering tools

To evaluate the performance of scMINER in cell clustering, we benchmarked it against five popular methods using 10 ground-truth scRNA-seq datasets (Supplementary Table 1). Each of these five methods utilizes one of three common clustering methodologies[18]: Seurat and Scanpy build nearest-neighbor graphs of cells in reduced dimensions and then detect clusters at specific resolutions; SC3s clusters cells using the consensus k-means algorithm; and scVI and scDeepCluster employ deep-learning models. The 10 datasets used for benchmarking were generated using seven common protocols for scRNA-seq profiling; these datasets varied in their feature counts, numbers of cells, and numbers of cell types.

After running each clustering method on each of the 10 datasets, we assessed how well each method recovered the ground-truth population labels using the Adjusted Rand Index (ARI), a widely used metric of cluster accuracy that assigns a value between 0 and 1, with values closer to 1 indicating greater accuracy[54]. scMINER produced the highest ARI in eight of the 10 ground-truth datasets and performed comparably to the top performers in the remaining two datasets (Fig. 2a). Overall, scMINER significantly outperformed the other methods, with an average ARI of 0.84 (Fig. 2b). In contrast to scMINER, the performance of the other methods was not consistent across datasets: for example, Seurat and Scanpy performed worse in smaller datasets (Yan, Buettner, Chung).

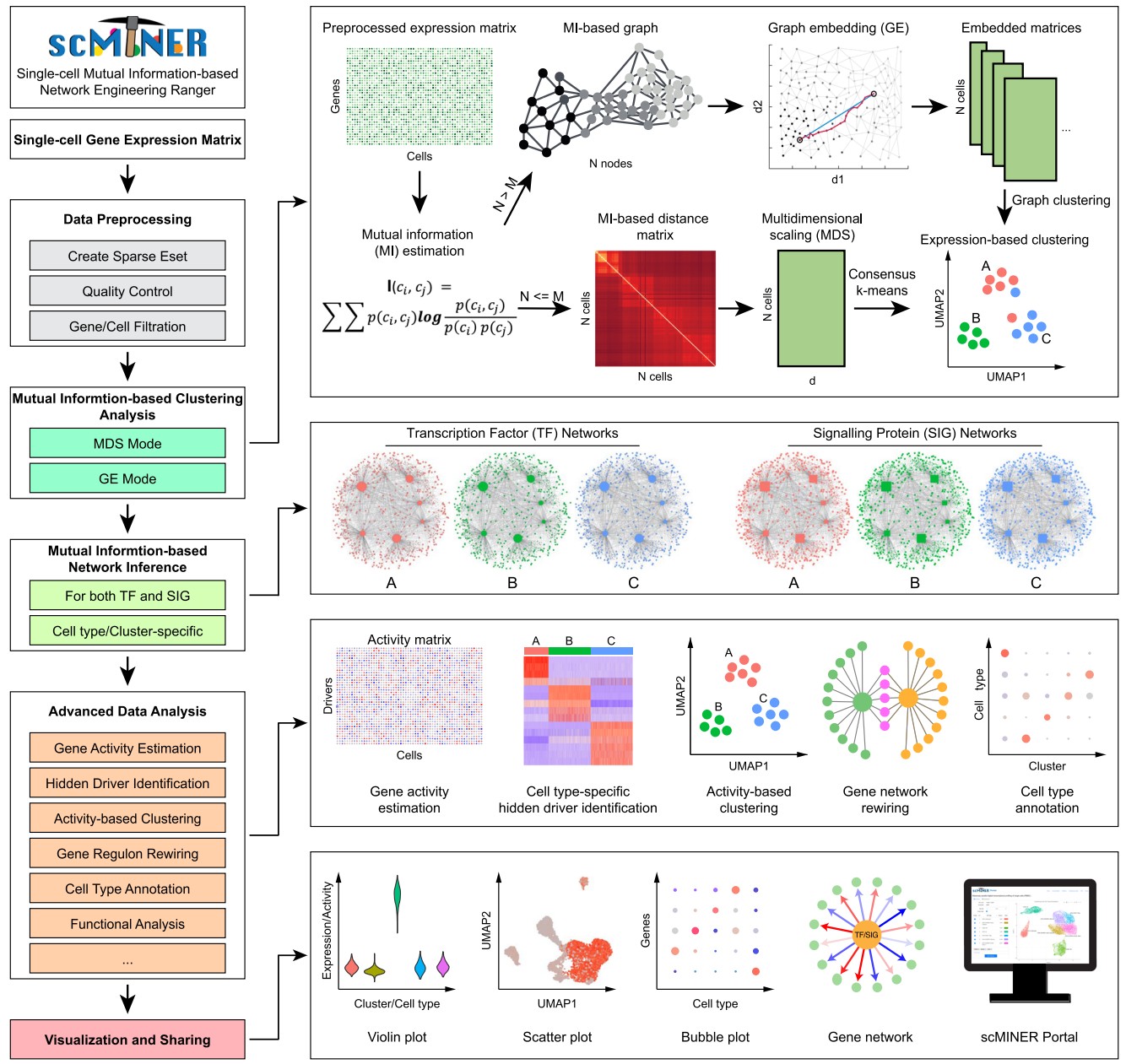

**Fig. 1 | The scMINER workflow.** scMINER is a mutual information-based systems biology framework designed for comprehensive end-to-end scRNA-seq data analysis. The workflow includes data quality control and filtration, MI-based accurate cell clustering, cell type-specific TF and SIG network inference, gene activity-based hidden driver identification, as well as data visualization and sharing.

To evaluate the accuracy and purity of the predicted cell clusters, we calculated the accuracy scores (percentage of major predicted labels in each ground-truth cluster) and purity scores (percentage of major ground-truth labels in each predicted cluster). Clusters generated by scMINER were the most accurate for seven and the purest for eight of the ten ground-truth datasets (Fig. 2c). Overall, scMINER produced the highest average accuracy and purity scores (Fig. 2d). scMINER also produced consistently better results as measured by Adjusted Mutual Information (AMI), Normalized Mutual Information (NMI), Average Silhouette Width (ASW) and Average Biological conservation score (AvgBIO) (Fig. 2e and Supplementary Fig. 2).

We continued our evaluation of scMINER by assessing its robustness vis-à-vis two parameters: the number of dimensions and bin size. scMINER primarily differs from its peers in that it utilizes MI-based cell-cell distance estimation and MDS-based dimension reduction. We therefore investigated how these metrics performed relative

to ones commonly used by existing single-cell clustering algorithms, using four gold-standard ground-truth datasets[55] (Yan, Pollen, Kolod, and Buettner). We first evaluated the effect of dimension size, a key factor of single cell clustering, on mutual information and three widely used cell-cell distance metrics: Euclidean, Pearson, and Spearman correlation coefficients. For this we replaced the default MI metric with these three, while keeping other steps of MICA workflow the same. Both MI and the other three distance metrics remained steady with five or more dimensions selected, with MI achieving higher ARIs than the others (Fig. 2f and Supplementary Fig. 3a), indicating that MI is both robust and accurate at capturing cell-cell similarity. We then benchmarked the effect of dimension size on MDS and three other common methods of dimensionality reduction: PCA, Laplacian Eigenmaps, and PCA and Laplacian performed in parallel (PCA/Laplacian) by changing the dimensionality reduction approach in MICA. The results obtained using MDS for dimensionality reduction increased in accuracy with

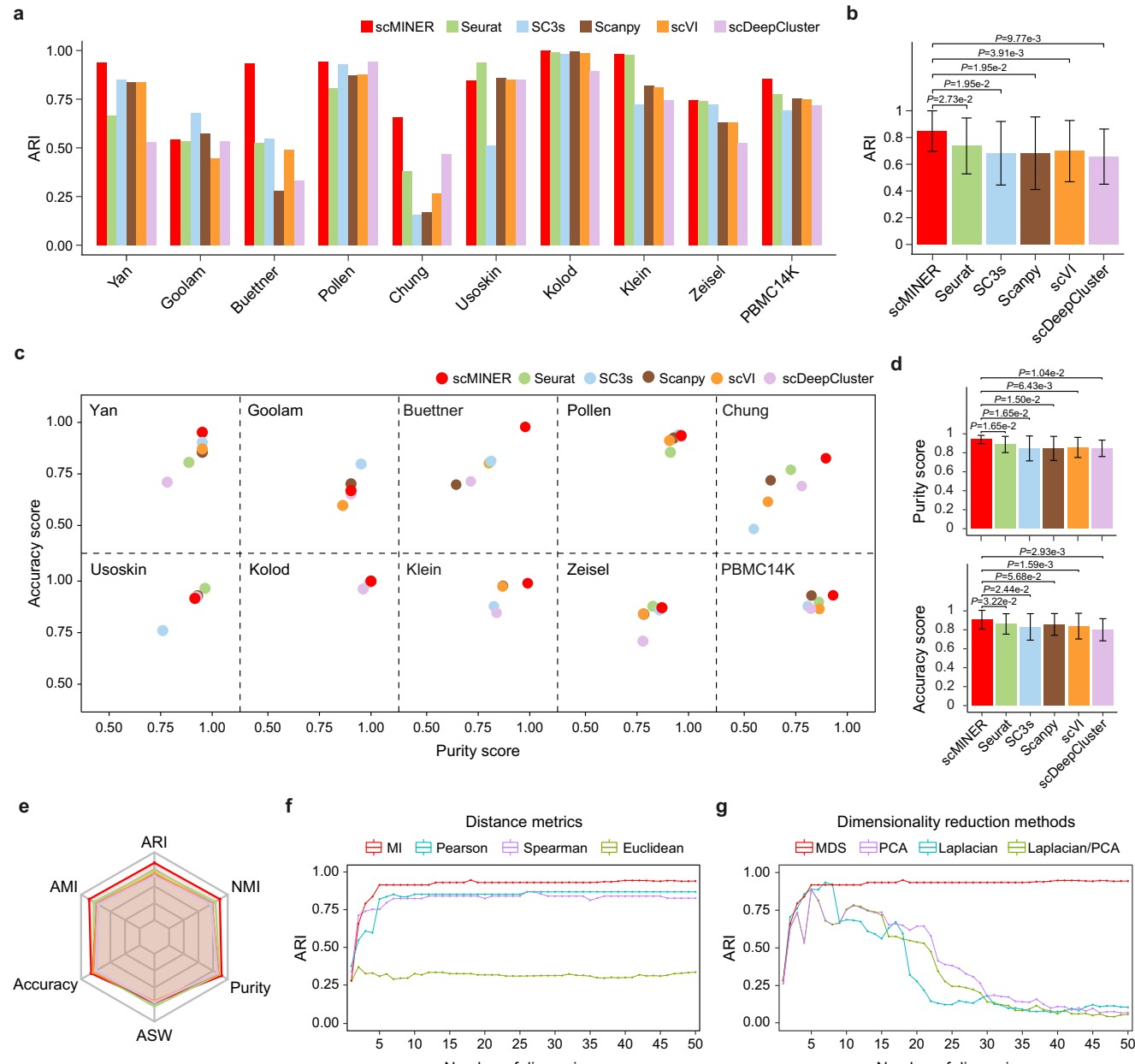

**Fig. 2 | scMINER provides accurate, robust, and scalable clustering for scRNA-seq analysis. a, b** Bar plots indicating the accuracy of clustering measured by ARI (Adjusted Rand Index) of scMINER and the other five benchmarked algorithms across 10 ground-truth datasets. Each bar in **b** represents the average ARI of each method across all datasets, and error bars represent the standard deviation. *P* value was estimated using paired samples Wilcoxon test, two-tailed. **c, d** Scatter plots (**c**) and bar plots (**d**) showing the accuracy and purity score of six benchmarked algorithms across 10 ground-truth datasets. Each bar in (**d**) represents the average purity (top) and accuracy (bottom) score of each method across all datasets, and

error bars represent the standard deviation. *P* value was estimated using the paired samples Wilcoxon test, two-tailed. **e** Radar plot summarizing the comparison between scMINER and the other five algorithms with six commonly-used cluster accuracy metrics. AMI: Adjusted Mutual Information; NMI: Normalized Mutual Information; ASW: Average Silhouette Width. **f, g** Line plots showing the effect of number of dimensions on cell-cell distance metrics (**f**) and dimensionality reduction methods (**g**) on clustering accuracy, respectively. The dots on each line indicate the average ARI of four ground-truth datasets of gold-standard, Yan, Pollen, Kolod, and Buettner. Source data are provided as a Source Data file.

increasing number of dimensions, topping out at five dimensions, at which point the accuracy remained stable as the number of dimensions continued to increase. This pattern did not hold when the other three approaches were used for dimensionality reduction. These approaches produced their most accurate results at different numbers of dimensions but failed to maintain that accuracy as the number of dimensions increased (Fig. 2g and Supplementary Fig. 3a). Notably, the default dimension size of 19 in MICA-MDS mode ensured the maximum accuracy across all benchmarked datasets. Furthermore, we benchmarked scMINER over the bin size, a known principal factor of

MI estimation, and revealed the high robustness of scMINER under different bin size from 5 to 100 across all benchmarked datasets (Supplementary Fig. 3b). Finally, we benchmarked the MICA-GE mode of scMINER, which is designated for large datasets, over dimension size and bin size using two larger datasets (Zeisel and PBMC14K). The results of both datasets highlighted that, for the MICA-GE mode, the ARI is primarily affected by Louvain resolution rather than the number of dimensions or bin size (Supplementary Fig. 3c, d).

We used the same 10 ground-truth datasets to benchmark scMINER against the five other clustering methods with regard to efficiency.

Five of the six methods, including scMINER, performed well in terms of running time (Supplementary Fig. 4a) and peak memory usage (Supplementary Fig. 4b). To compare the scalability of these tools, we introduced three atlas-scale datasets, HMC76K, Covid97K and Covid650K, into the efficiency benchmarks. All six methods successfully completed the clustering analysis of these three large-scale datasets (Supplementary Fig. 5a) and scMINER captured the major structure of each dataset (Supplementary Fig. 5b). Consistent with the results obtained using the original 10 datasets, scDeepCluster had a significantly longer run time and greater peak memory usage compared to the other methods. Though quantifying cell-cell distance using MI that has been calculated from all genes is computationally demanding, scMINER's fast MI estimation and use of optimized cell-cell nearest neighbor graph building allows it to process large datasets with run times and peak memory consumption comparable to other methods (Supplementary Fig. 4c). Altogether, our systematic, comprehensive benchmarking demonstrated that scMINER is a highly accurate, robust, and scalable method of scRNA-seq data clustering analysis.

## scMINER improves the clustering of ambiguous populations

To evaluate the clustering performance of scMINER on a well-characterized mixed cell population, we compiled a dataset of 14,000 peripheral blood mononuclear cells (PBMCs) comprising seven mutually exclusive cell types from the Zheng dataset[56]. While all benchmarked methods were able to distinguish monocytes, B cells, and natural killer (NK) cells from T cell populations (Fig. 3a and Supplementary Fig. 6a, b), scMINER was the only method to distinguish between CD4+ regulatory T cells (CD4 Treg) and CD4+ central memory T cells (CD4 TCM): CD4 Treg and CD4 TCM cells were well separated by scMINER but had major overlaps by the other methods as visualized in the UMAP embeddings (Fig. 3b). Having identified scMINER and Seurat as the top performers in our previous benchmarking experiment, we gave special attention to their respective ability to distinguish T cell subpopulations. We first assessed the purity of the clusters of T cell populations predicted by scMINER and Seurat, respectively. scMINER and Seurat both captured CD8+ Naïve T cells (CD8 TN) with over 90% purity and CD4+ naïve T cells (CD4 TN) with approximately 80% purity (Fig. 3c, d). However, scMINER produced purer clusters for CD4 Treg and CD4 TCM (96% purity to Seurat's 78% and 92% purity to Seurat's 61%, respectively). We then evaluated the accuracy of these clusters and found that scMINER better captured true CD4 Treg and CD4 TCM subpopulations than Seurat (70% accuracy to Seurat's 52% and 95% accuracy to Seurat's 88%, respectively) (Fig. 3e, f). The other benchmarking methods likewise struggled to distinguish T cell populations (Supplementary Fig. 6c, d). Overall, scMINER was the best at distinguishing ambiguous T cell subpopulations and generating biologically meaningful clusters for downstream analysis of the PBMC14K dataset.

To determine whether the selection of 2,000 highly variable genes (HVGs) limits other five methods' ability to differentiate between T cell subpopulations in the PBMC14K dataset, we repeated the analysis with these methods using 1,000 and 3,000 HVGs. Changing the number of HVGs used for cell clustering had no significant effect on clustering accuracy (Supplementary Fig. 7a), nor did it affect the methods' ability to distinguish between CD4 Treg and CD4 TCM cells (Supplementary Fig. 7b), with one exception: Scanpy's ARI increased from 0.69 using 1,000 HVGs to 0.82 using 3,000 HVGs. While increasing the number of HVG's improved Scanpy's performance, its ARI was still lower than scMINER's. We also examined the clustering performance of scMINER and Seurat at different Louvain resolutions, with the number of inferred clusters, k, ranging from 4 to 10. At each resolution, scMINER outperformed Seurat, producing a higher ARI (Supplementary Fig. 7c). At lower resolutions, both scMINER and Seurat can distinguish B cells, monocytes, NK cells and T cells. At higher clustering resolutions, scMINER begins to separate T cells into

four subpopulations, but Seurat's clustering of T cell subpopulations remains unchanged (Supplementary Fig. 7d, e). All in all, scMINER clusters ambiguous T cell subpopulations better than leading cell clustering methods, making it an outstanding tool for uncovering cell-type-specific biology.

## scMINER enables accurate TF network inference

scMINER employs SJARACNe[46], a mutual information-based computational framework for network inference from single-cell data. To tackle the sparsity of scRNA-seq data, the user can optionally generate pseudo-bulk aggregates using SuperCell[47] before network reconstruction. Hereby, we evaluated and found that scMINER outperforms three existing methods, GENIE3[3], GRNBoost2[5], and PIDC[11], in terms of genome-scale TF network reconstruction using multiple datasets and evaluations.

In the first evaluation, we coupled scRNA-seq data, which was used for network inference, with bulk ATAC-seq data, which was used to reconstruct the ground-truth TF network by motif footprinting. These data included three subtypes of exhausted CD8+ T cells[57]: progenitor exhausted CD8+ T (Tpex) cells, effector-like CD8+ T (Teff-like) cells, and terminally exhausted CD8+ T (Tex) cells and two types of tissue-specific regulatory T cells from mouse spleen (Tregs Spleen) and visceral adipose tissue (Tregs VAT)[58] (Supplementary Table 2). First, we evaluated the accuracy of the global reconstructed TF networks based on the number of true positive predictions as a function of top-ranked edges, early precision ratio[17] and compared receiver operating (ROC) and principal response curve (PRC) characteristics (Fig. 4a for Teff-like cells; Supplementary Fig. 8a for Tpex, Tex, Tregs Spleen, and Tregs VAT). scMINER exhibited the largest number of true positive predictions among top-ranked edges and the highest early precision ratio. Also, scMINER outperformed alternative methods based on the area under the ROC (AUROC) and PRC (AUPRC) curves (Fig. 4a and Supplementary Figs. 8a, 9).

We subsequently evaluated the accuracy of predicted targets for individual TFs. In Tpex, Spleen, and VAT Tregs cases, the network predicted by scMINER exhibited a greater AUPRC ratio than the networks predicted by other methods. In Teff-like and Tex cases, the scMINER-predicted networks had AUPRC ratios that were greater than the PIDC- and GENIE3-predicted networks' and roughly equivalent to GRNBoost2's (Fig. 4b and Supplementary Fig. 8a). We observed similar results (increased performance or no significant difference) by using alternative evaluation metrics, like AUROC or AUPRC ratio for a full list of reported targets (Supplementary Fig. 9).

While ATAC-seq can potentially validate the TF-target binding interactions, it does not provide information on the functional consequences of the interaction. To address this issue, we performed the second evaluation using Perturb-seq, which can capture cell-type-specific up or down-regulated genes upon perturbation of multiple TF or other genes in parallel by coupling CRISPR and scRNA-seq technologies[59,60]. Specifically, we predicted TF networks using scMINER and other three methods from scRNA-seq profiles of cells with non-targeting sgRNA (NT) only and compared them with the ground-truth TF networks reconstructed by performing differential expression analysis of scRNA-seq profiles of cells with CRISPR knockout (CRISPR-KO), interference (CRISPRi), or activation (CRISPRa) vs. NT perturbations for each TF. ROC analysis for the global TF network inferred from a CRISPR-KO Perturb-seq dataset[59] (Fig. 4c) showed that scMINER network demonstrated a higher AUROC value (0.69) than those predicted by GENIE3 (0.65), GRNBoost2 (0.57) and PIDC (0.52). We subsequently evaluated the predicted networks at the level of individual TFs using gene set enrichment analysis (GSEA) with the log2 fold-change of CRISPR-KO vs. NT as the reference. Overall, scMINER exhibited significantly stronger enrichment of predicted targets for 96 TFs than alternative methods (Fig. 4d, e). Taking EGR2 as an example, we predicted 444 targets, 41 of which exhibited significant differential

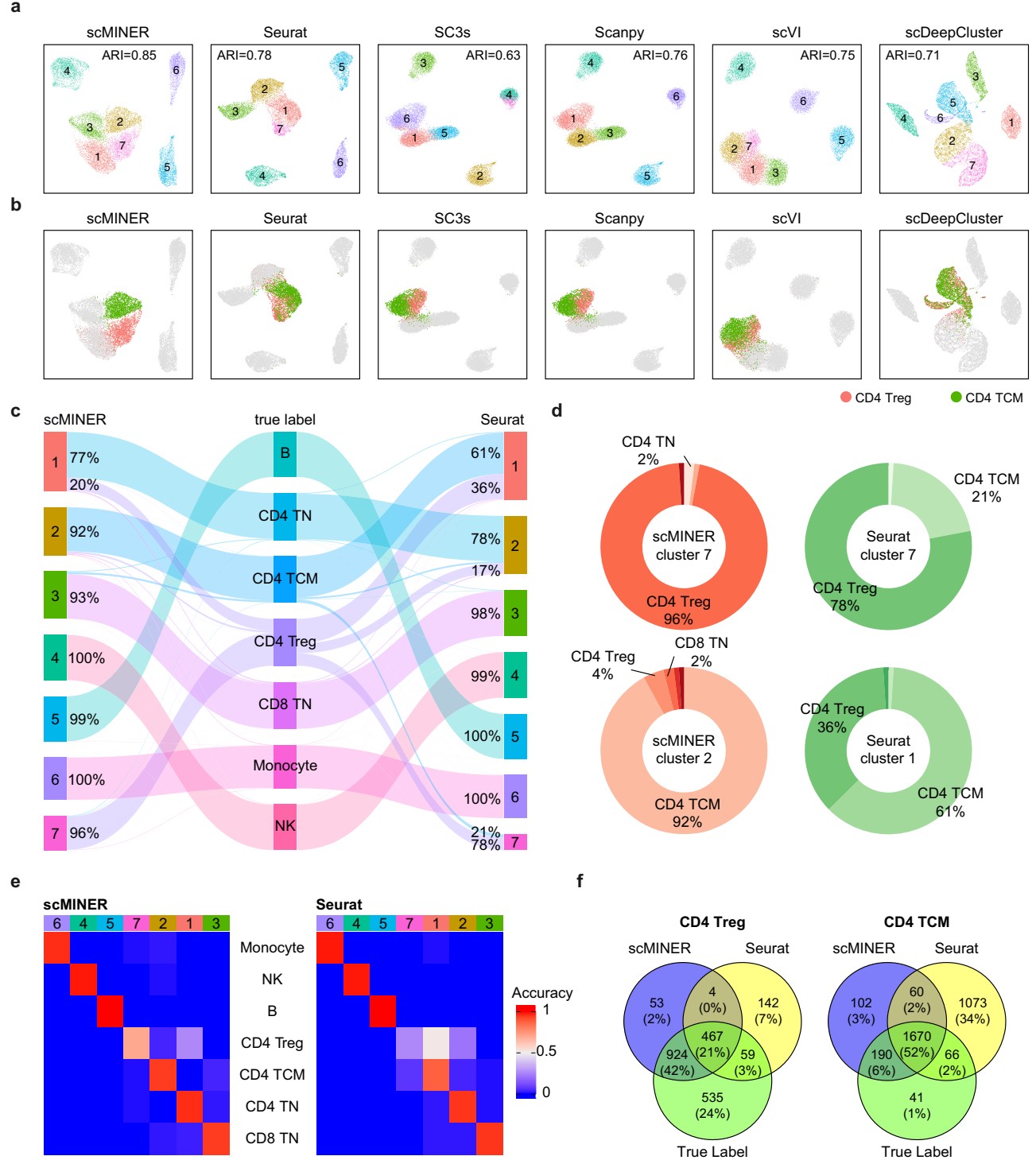

**Fig. 3 | scMINER improves the clustering of ambiguous subpopulations in PBMCs. a, b** UMAPs of cell clusters predicted by scMINER and other five benchmarked algorithms (**a**). The projections of CD4 Treg and CD4 TCM cells are highlighted based on ground-truth labels (**b**). **c** Sankey plot of the overlap between the ground-truth labels (middle) and the labels predicted by scMINER (left) and Seurat (right), respectively. **d** Donut plots showing the cell composition of major clusters predicted as CD4 Treg (top) and CD4 TCM cells (bottom) by scMINER (left) and Seurat (right), respectively. **e** Heatmaps comparing the accuracy of clusters predicted by scMINER (left) and Seurat (right), respectively. **f** Venn diagrams summarizing the overlap between ground-truth labels and the labels predicted by scMINER and Seurat for CD4 Treg (left) and CD4 TCM cells (right). CD4 Treg: CD4+ regulatory T cells; CD4 TCM: CD4+ central memory T cells; CD4 TN: CD4+ naïve T cells; CD8 TN: CD8+ naïve T cells; NK: natural killer cells. Source data are provided as a Source Data file.

expression in Perturb-seq data with a cut-off of |log2FC| > 0.25 (Fig. 4f). scMINER demonstrated a lower P-value (5.6e-52) than the other three methods (5.7e-25 for GENIE3, 4.3e-25 for GRNBoost2 and 4.1e-24 for PIDC), which was also consistent with the ROC analysis (AUR-OC.scMINER=0.77, AUROC.GENIE3 = 0.58, AUROC.GRNBoost2 = 0.55,

AUROC.PIDC = 0.51) (Fig. 4g). A unique feature of scMINER, compared to the other methods used in this evaluation, is that scMINER can distinguish positive and negative targets for each TF. As demonstrated by GSEA, we observed a significant positive enrichment (P = 3.15e-23) of negative targets and a negative enrichment (P = 1.43e-31) of positive

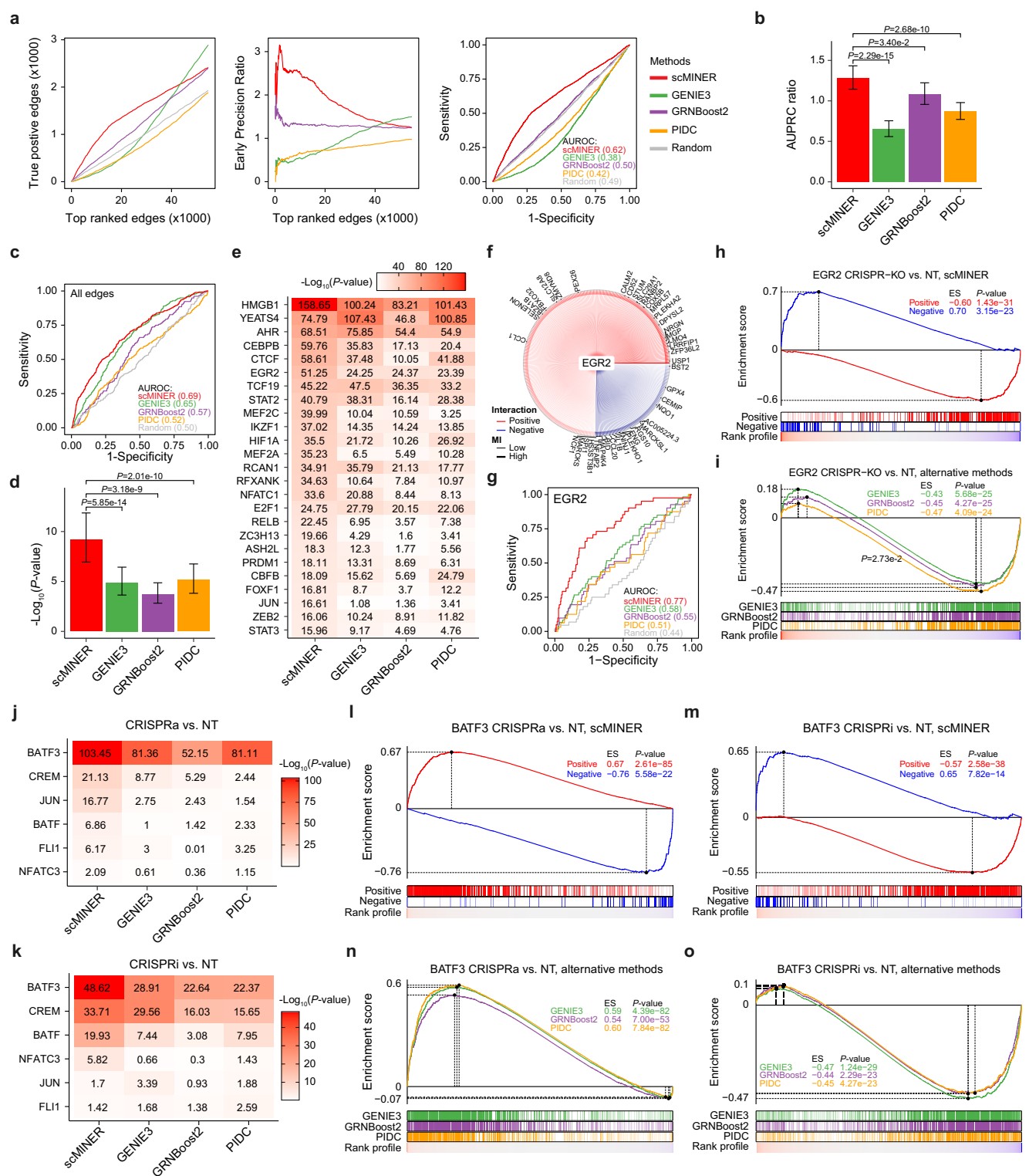

targets (Fig. 4h), while the other three methods mixed all targets in the case of EGR2 (Fig. 4i).

For further validation, we employed an orthogonal CRISPRa- and CRISPRi-based Perturb-seq dataset[60] in which both CRISPRa and CRISPRi perturbations were available for each of 6 TFs. Similar to the previous CRISPR-KO evaluation, we observed significantly higher GSEA scores for targets predicted by scMINER than for the three reference methods (Fig. 4j, k). The matched CRISPRa and CRISPRi could be used to cross validate scMINER's ability to predict the direction of targets. For example, in the network predicted by scMINER for BATF3, the

positive targets were positively enriched for BATF3 CRISPRa vs. NT and negatively enriched for BATF3 CRISPRi vs. NT, while negative targets followed the opposite pattern (Fig. 4l, m). Other methods used in our evaluation did not distinguish between positive and negative targets and exhibited lower enrichment scores (Fig. 4n, o). In summary, our evaluation using Perturb-seq datasets allows us to conclude that the accuracy of scMINER in the reconstruction of TF networks exceeds one of the widely used methods like GENIE3, GRNBoost2 and PIDC. Also, scMINER provides information about the positive and negative nature of regulation, which can be critical in biomedical applications.

**Fig. 4 | scMINER improves the accuracy of transcription factor network inference.** Benchmarking of TF network inference by scMINER and three other algorithms using the ground-truth networks defined by ATAC-seq (**a, b**) and CRISPR perturbation datasets (**c–o**). **a** Network-level evaluations using three metrics: number of true positive predictions (left), early precision ratio (middle), and ROC curves of top-ranked edges (n = 54,800) predicted by each method. **b** Regulon-level evaluation using AUPRC ratios of predicted TF targets. Each bar represents the median AUPRC ratio across all regulons, and error bars represent the 95% confidence intervals. *P*-values were estimated using the paired Wilcoxon rank sum test. AUPRC: area under the precision-recall curve. **c** ROC curves evaluating TF networks inferred by each method. True labels were identified based on differential expression analysis. AUROC: area under the receiver operating characteristic curve. **d** Bar plots indicating the $-\log_{10}(P\text{-values})$ computed with Kolmogorov–Smirnov test of GSEA for all TF genes evaluated with Perturb-Seq. Each bar indicates the median value, and error bars represent 95% confidence interval. P-values were computed with the paired Wilcoxon rank sum test. **e** Heatmap summarizing the $-\log_{10}(P\text{-values})$ of GSEA for the top 25 predicted TF genes. **f** The regulon of EGR2 predicted by scMINER (n = 444). Red and blue edges indicate positive and negative targets, respectively. Symbols of the target genes with $|\log_2 FC| > 0.25$ are shown (n = 41). **g** ROC curves of EGR2 target genes predicted by each method. **h, i** Enrichment plots of EGR2 target genes predicted by scMINER (**h**) and the other three methods (**i**). P-values were calculated by Kolmogorov-Smirnov test of GSEA. Genes were ranked by the $\log_2 FC$ of CRISPR-KO (higher) vs. NT (lower). **j, k** Heatmap summarizing the $-\log_{10}(P\text{-values})$ of GSEA for CRISPR activation (**j**) and interference (**k**). **l–o** Enrichment plots of BATF3 target genes predicted by scMINER (**l, m**) and the other three methods (**n, o**) for both CRISPR activation (**l, n**) and interference (**m, o**) screens. The enrichment scores and P values were calculated by Kolmogorov-Smirnov test of GSEA. Genes were ranked by the $\log_2 FC$ of CRISPRa/i (higher) vs. NT (lower). CRISPRa: CRISPR activation; CRISPRi: CRISPR interference. Source data are provided as a Source Data file.

Finally, we evaluated scMINER and other algorithms by examining to what extent their inferred networks can recapitulate 18 well-known TF regulators of three subsets of exhausted T cells[57]. Specifically, we used exhausted T cell networks reconstructed with each method and computed the activity of each known TF driver across the three exhausted T cell subsets (see Methods). We then predicted the substate for each TF based on the TF activity computed from the networks inferred by each method and evaluated how well they agreed with prior biological knowledge. Out of 18 known TF drivers, scMINER correctly predicted 17, achieving the highest prediction rate of 94%, compared to 78% for PIDC, 39% for GRNIE3, and 28% for GRNBoost2 (Supplementary Fig. 8b). These results showed that scMINER-reconstructed TF network allows for accurate estimation of driver activity which is essential for novel driver inference and functional characterization of biomedical data. Collectively, these evaluation results demonstrated that scMINER consistently outperforms existing popular algorithms for TF regulatory network inference.

## scMINER outperforms SCENIC in TF driver identification

Accurate reconstruction of TF regulatory network from scRNA-seq data by scMINER enables us to comprehensively infer TF activity and identify cell type-specific TF drivers. For this application, we benchmarked scMINER against SCENIC[12], the most popular TF driver inference algorithm, using three datasets: PBMC14K, exhausted T cells, and tissue-specific Tregs used in previous clustering or network inference evaluations. We compared scMINER-derived activity against gene expression and SCENIC-derived activity on how well they captured literature-reported TF drivers in each of the three datasets. In the PBMC14K dataset, scMINER demonstrated a superior performance in characterizing the activities of 14 positive control TF drivers across different immune cell types (Fig. 5a and Supplementary Table 3). All three methods (expression, SCENIC, scMINER) were able to capture most TF drivers of non-T cell populations, such as CEBPA[61], CEBPB[61], SPI1[62] in monocytes, IRF8[63] in B cells, and TBX21[64] and PRDM1[65] in NK cells. Notably, the activities generated by scMINER and SCENIC for these markers both produced clearer and more well-defined signal than the gene expression alone. For T cell subpopulations, the expression data along and the activity computed by SCENIC, however, produced signal that was much more ambiguous, making the activity produced by scMINER the only method that can discern them (Fig. 5a). For instance, the expression data for FOXP3, a marker of CD4$^+$ regulatory T (CD4Treg) cells, was very sparse; the activity reported for FOXP3 by SCENIC, though less sparse, did not show clear signal; the activity generated by scMINER, meanwhile, produced clear signal from which a CD4Treg population could be identified (Fig. 5b). Besides producing clearer signal, scMINER computed the activity for a significantly greater number of genes (850 TFs) than SCENIC (297 TFs) which missed some important TFs like SATB1 (Fig. 5a).

Following the practice in SCENIC, we next used the activity values estimated by scMINER to perform cell clustering and benchmarked against SCENIC. The Louvain clustering with the same parameters for both activity inputs showed that both scMINER and SCENIC were able to clearly distinguish between monocyte, B, and NK populations (Fig. 5c, d). For T cell populations, SCENIC produced more ambiguous results; as a result, clustering based on the activity reported by SCENIC had a notably lower ARI value (0.69) than clustering based on the activity reported by scMINER (0.87, Fig. 5e). Intriguingly, the ARI of activity-based cell clustering is slightly higher than that using expression data only (ARI = 0.85, Fig. 3a).

We then repeated the workflow used previously in the PBMC14K dataset with the exhausted T cell dataset[57]. As previously, the scMINER-estimated activities of 18 curated TF drivers of T cell exhaustion stages (Supplementary Table 3) produced clearer and more specific signals than did the expression data only and SCENIC-derived activities (Fig. 5f). It is worth noting that SCENIC failed to generate activity for four well-defined markers, including Tcf7[66] and Id3[66] for Tpex, and Batf[67] and Tox[68] for Tex. Among the 14 TF drivers for which SCENIC did generate activity, the signal of some drivers was less well defined, such as Rel[69] and Nfkb1[69] for Tpex, and E2f2[70] and Bhlhe40[71] for Tex. In contrast, scMINER computed activity for all 18 TF drivers, and the signal for that activity was clear-cut (Fig. 5f and Supplementary Fig. 10a, b). Consider Batf: no clear pattern emerged from its expression, which was distributed evenly across all three subpopulations. SCENIC failed to capture Batf activity wholesale. scMINER, on the other hand, not only captured the Batf activity but also successfully pinpointed that activity to the Tex population (Fig. 5g). Consistent with the findings in the PBMC14K dataset, the activity generated by scMINER produced better clustering results than SCENIC as well (Fig. 5h, i).

Notably, as the scMINER workflow features cell type-specific network inference, scMINER is able to unearth TF regulon rewiring among different cell populations, enabling the identification of the mechanisms underlying cell state transitions. Indeed, scMINER network analysis revealed significant rewiring of Batf regulon targets among Tpex, effector-like Tex, and terminal Tex states (Fig. 5j, k), consistent with a recent study reporting that BATF promotes the transition from Tpex to Teff-like cells[72].

The patterns observed in the previous two datasets were generally preserved in the tissue-specific regulatory T-cell dataset[58]. Signals for literature-curated TF drivers were again clearer and more well-defined in the activity generated by scMINER (Supplementary Fig. 11a and Supplementary Fig. 10c, d). The example of Pparg (a known TF master regulator of VAT Tregs) showed that activity signal computed with scMINER could compensate for the sparsity of single-cell expression data and produce a more specific signal in VAT Tregs than SCENIC (Supplementary Fig. 11b). Clustering analysis based on

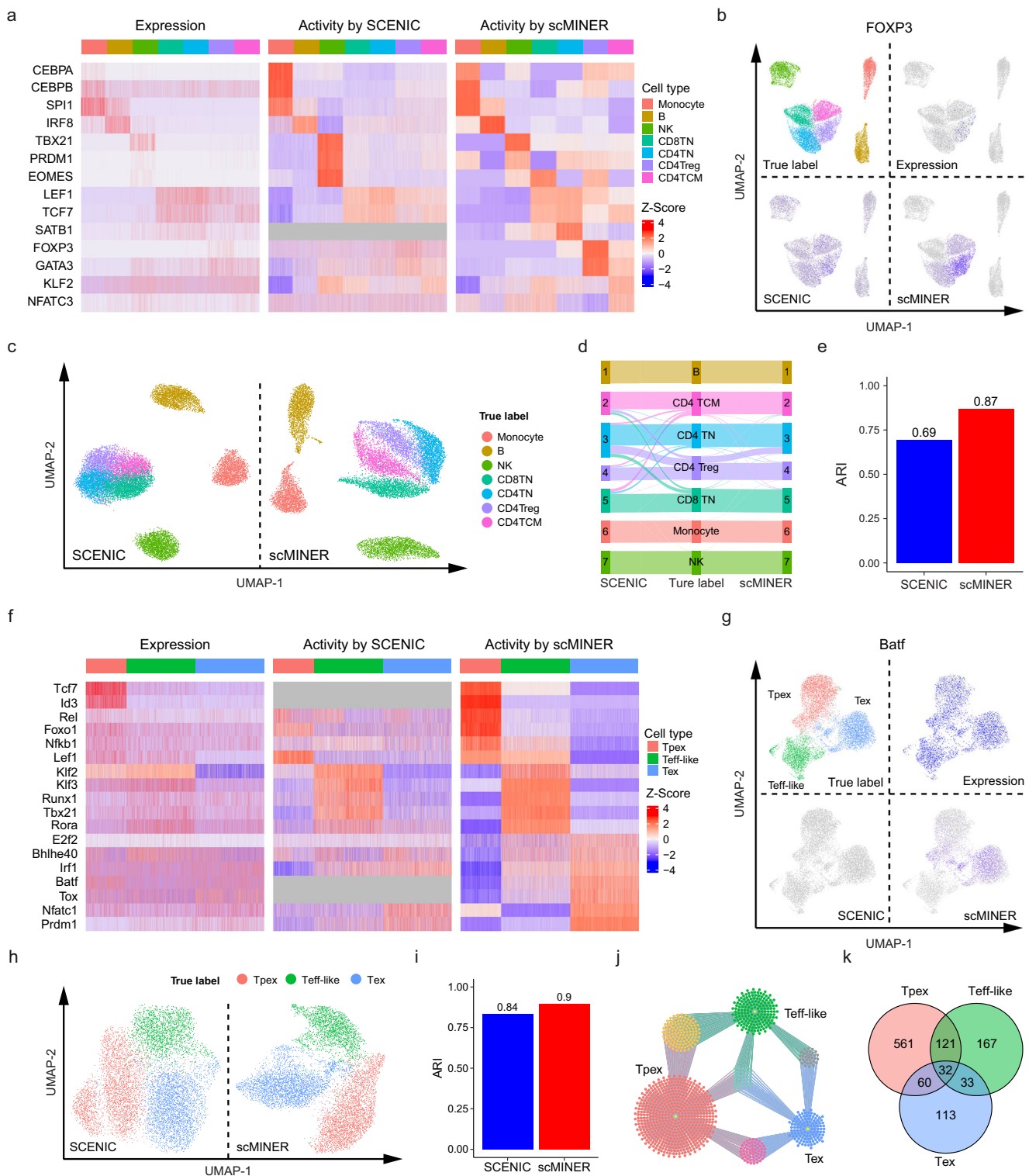

scMINER-inferred activity achieved an almost perfect ARI (0.998) despite a small increase compared with SCENIC likely due to easy separation of the four cell populations (Supplementary Fig. 11c, d). We also observed a significant rewiring of Pparg targets among tissue-specific Tregs (Supplementary Fig. 11e), suggesting its different functions underlying Treg tissue specification.

Altogether, the results in these three datasets demonstrate that scMINER consistently outperforms SCENIC in both TF driver identification and activity-based cell clustering. These findings support the superiority of scMINER's estimation of TF driver activity.

## scMINER accurately infers SIG networks and drivers

Signaling proteins, like transcription factors, have been recognized as essential regulators of a variety of biological processes and catastrophic diseases[40,41]; they are also more druggable than TFs[42,43]. However, few methods exist to infer SIG networks from scRNA-seq data, making it challenging to identify SIG drivers. To overcome this limitation, we added signaling network reconstruction and activity analysis to the scMINER workflow.

To assess the accuracy of SIG network reconstruction by scMINER, we applied a similar approach as in the case of TF network by using

**Fig. 5 | scMINER outperforms SCENIC in TF driver identification.** Results of performance benchmarking of activity-based analysis by scMINER and SCENIC in two different datasets: PBMC14K (**a**–**e**, *n* = 13,605 total cells from 7 cell types) and exhausted T cells (**f**–**k**, *n* = 8464 total cells from 3 CD8 + T cell linages). **a** Heatmaps showing the expression (left), activity computed with SCENIC (middle), and activity computed with scMINER (right) for cell type-specific TF markers of PBMCs. **b** UMAPs of PBMC dataset highlighting the true cell type labels (top-left), FOXP3 expression (top-right), and FOXP3 activity computed with SCENIC (bottom-left) and scMINER (bottom-right). **c** UMAPs of activity-based clustering by SCENIC (left) and scMINER (right). **d** Sanky plot of the overlap between the ground-truth labels and the labels predicted by activity-based clustering with SCENIC and scMINER, respectively. **e** ARI values for clustering based on the activity generated by SCENIC- and scMINER, respectively. **f** Heatmaps showing the expression (left), activity computed with SCENIC (middle), and activity computed with scMINER (right) for cell type-specific markers of exhausted T cells. **g** UMAPs of exhausted T cells highlighting the true cell type labels (top-left), Batf expression (top-right), Batf activity computed with SCENIC (bottom-left), and with scMINER (bottom-right). **h** UMAPs of activity-based clustering by SCENIC and scMINER. **i** ARI values of clustering based on the activities generated by SCENIC and scMINER, respectively. **j** Regulon rewiring diagram of Batf among Tpex (red), Teff-like (green), and Tex (blue) cells. The genes shared by two or more regulons are clustered and color-coded (yellow for Tpex and Teff-like, gray for Teff-like and Tex, magenta for Tex and Tpex). **k** Venn plot showing the overlap of Batf regulons among Tpex, Teff-like, and Tex cells. ARI Adjusted Rand Index, Tpex progenitor exhausted T cells, Teff-like effector-like T cells, Tex terminally exhausted T cells (Tex). Source data are provided as a Source Data file.

Perturb-seq. Using a Perturb-seq dataset of immune response with CRISPR knock-out of SIG genes[46], we built a ground-truth network of 294 SIG factors by performing differential expression analysis as previously described and used it to evaluate the scMINER-predicted SIG network from scRNA-seq profiles of control cells with non-targeting perturbations. For global SIG network evaluation, we performed true positive (TP), early precision ratio (EPR) and ROC analyses, demonstrating that scMINER, with a AUROC of 0.75, performs significantly better that random predictor in retrieving the targets of SIGs defined by CROP-seq data (Fig. 6a). For individual SIG network evaluation, we performed both GSEA and ROC analyses as we did for TF network evaluation. Out of 294 SIG factors, 207 showed significant enrichment of scMINER-inferred SIG regulons in differentially expressed genes from Perturb-seq data, and most exhibited greater AUROC values compared to random (Fig. 6b). Taking IFNGR2 as an example, out of 188 targets predicted with scMINER, 25 showed significant differential expression in Perturb-seq data with a cutoff of |log2FC| > 0.25 (Fig. 6c). The GSEA results showed that scMINER-inferred positive targets of IFNGR2 were significantly enriched in down-regulated genes upon CRISPR-KO while predicted positive targets were enriched in the opposite direction (Fig. 6d), consistent with the high AUROC value of 0.81 by ROC analysis (Fig. 6e). The Perturb-seq evaluation suggested that scMINER was able to predict cell type specific SIG networks with direction from scRNA-seq data with reasonable accuracies.

Downstream of SIG network inference in scMINER was to infer cell-type-specific SIG drivers. To validate the accuracy of scMINER-predicted SIG drivers, we used CITE-seq, a sequencing-based method that simultaneously quantifies cell surface proteins and transcriptomes within a single cell[73]. To do so, we selected two CITE-seq datasets of human PBMCs that were generated through different protocols[74,75]; estimated the activities of all captured cell surface proteins using scMINER; and then used the protein measurements to identify the marker genes for each cell type. While both the activity and expression measurements recapitulated most of the previously identified markers, the activity measurement had a comparatively higher specificity and was less affected by dropout (Fig. 6f and Supplementary Fig. 12a). Notably, the activity measurement better distinguished ambiguous CD4+ and CD8+ T cell populations than did the expression and protein measurements (Fig. 6f). The activity measurement also captured the known markers of each cell type with greater specificity (Fig. 6g, h and Supplementary Fig. 12b, c).

Finally, similar to TF activity estimation based on inferred network, scMINER can quantify the activity of SIG drivers. We applied our workflow to analyze signaling activity in PBMC, exhausted T-cell, and tissue-specific regulatory T-cell datasets from the previous sections and then compared the expression and activity of known signaling marker genes (Supplementary Fig. 13). The activity generated by scMINER was able to compensate for the signal drop-off in the expression data, significantly increasing the specificity of the signal. For instance, in the PBMC dataset, signal for the expression of NCAM1, a marker of NK cells, is very sparse; however, the activity generated by scMINER makes it possible to identify NK cells using that marker (Supplementary Fig. 13a, b). Similarly, in the exhausted T-cell set, the signal for expression of Prdm1 is very sparse and non-specific, spanning Tpex and Teff-like populations; the activity generated by scMINER, meanwhile, was limited to the Tex population, consistent with a study tracing Prdm1 to Tex cells (Supplementary Fig. 13c, d). Similar results were observed in the tissue-specific regulatory T cells, as exemplified by the case of the Hells gene (Supplementary Fig. 13e, f).

## Discussion

In this study, we present scMINER, a novel computational framework that leverages mutual information to analyze scRNA-seq data, allowing for the quantification of complex dependencies between cells and genes. Through benchmarking with datasets containing ground-truth labels, we demonstrate that scMINER achieves higher accuracy in clustering of scRNA-seq data and distinguishes ambiguous cell populations more effectively than current state-of-the-art methods. Furthermore, we show that scMINER surpasses existing algorithms (GENIE3, GRNBoost2, PIDC) in TF network inference and outperforms SCENIC in TF driver prediction, supported by data from ATAC-seq and Perturb-seq. Notably, scMINER is the only method capable of inferring SIG networks and drivers supported by data from both Perturb-seq and CITE-seq. Additionally, scMINER adeptly predicts both positive and negative targets of a TF or SIG, as confirmed by Perturb-seq, and elucidates their rewiring across different cell types or states. Moreover, we provide a user-friendly interactive web portal, enabling researchers to effortlessly explore scMINER's outputs in cell clustering, activity estimation, and network inference.

The superior performance of scMINER owes to its several elements: first, its use of mutual information captures both linear and non-linear dependencies among genes and cells; second, scMINER's re-engineering of SJARACNe and utilization of a MetaCell strategy mitigates dropout effects and reduces the false-positive rate; finally, scMINER's inclusion of SIG network inference significantly enhances its ability to identify hidden drivers by incorporating more than 5,500 SIG proteins in addition to the approximately 1,500 TFs that form the basis for most existing methodologies.

scMINER's inference of cell type-specific gene networks and identification of hidden drivers hold great promise for mechanistic studies of disease and the discovery of therapeutic targets. For instance, we have demonstrated that scMINER is able to capture activities of FOXP3[76] in Tregs and BATF[67] in T cell exhaustion, for which scRNA-seq falls short in identification and SCENIC fails entirely to identify. We have also demonstrated that scMINER effectively captures CD4 activity in CD4+ T cells while scRNA-seq or CITE-seq have limited power. The utility of scMINER is further emphasized by its ability to detect EOMES activity in NK, CD8 and CD4Tregs while SCENIC is limited to capturing its activity only in NK cells[64,77]. Further, our most recent study[78] used the scMINER-generated single-cell networks to delineate the gene activity landscape of B cell development and identified BCL2 as a driver of asparaginase resistance in this population with validation both in vitro and in vivo.

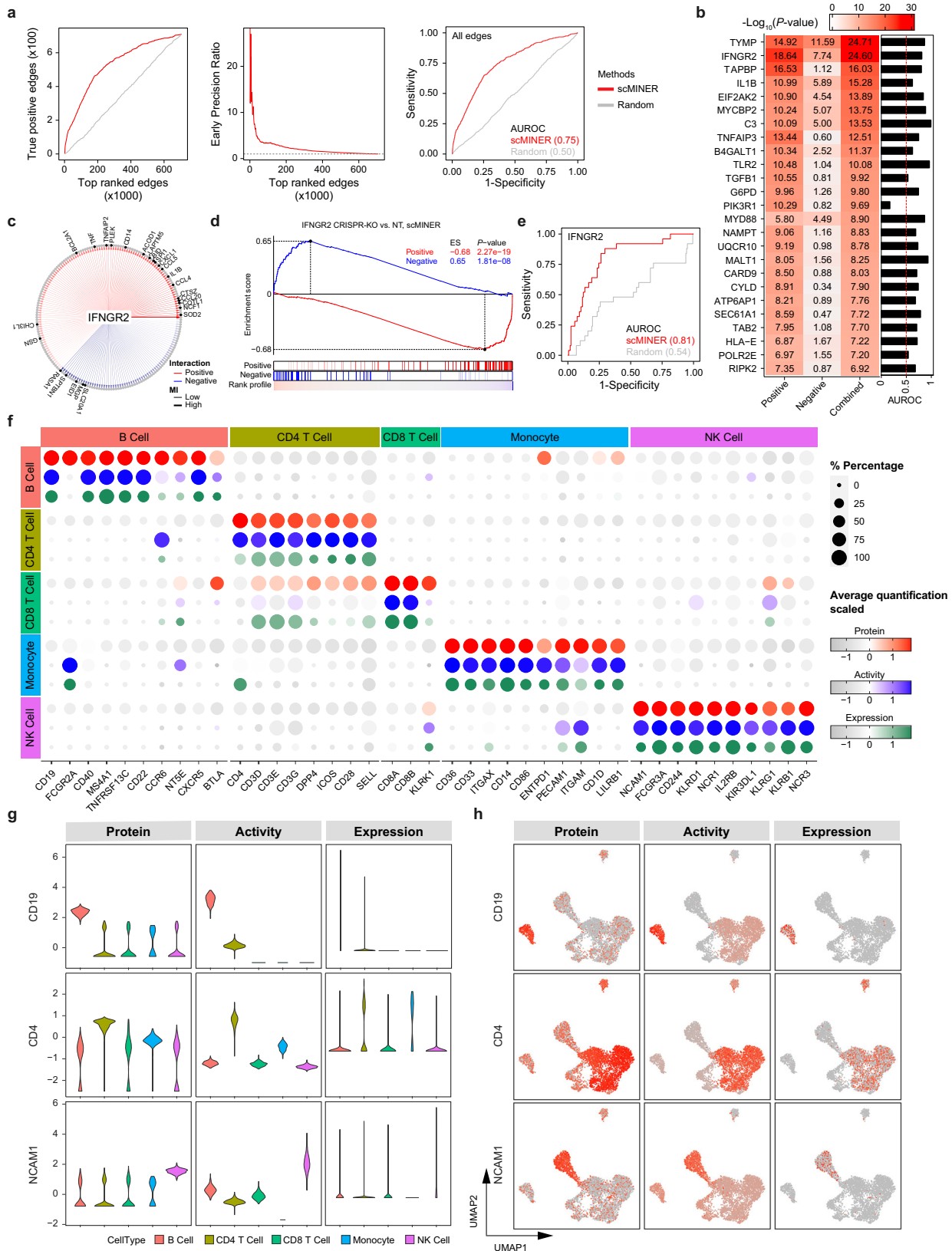

While scMINER is adept at reconstructing networks and inferring TFs and SIGs driver from scRNA-seq data, it comes with some limitations. scMINER requires a minimum of 500 cells, limiting its applicability in studies involving rare cell populations. Various methods, both computational[79–81] and experimental[82], have been developed for amplifying the signal of rare cell populations; these approaches could

potentially be deployed prior to applying scMINER, but the degree to which they would benefit scMINER's performance remains to be seen. Another potential way to circumvent this problem would be to use the publicly available data of same biological contexts than can be found by mapping these rare cell populations to reference databases (e.g., the Human Cell Atlas[83]); this approach holds promise given the rapid

**Fig. 6 | scMINER enables the accurate identification of SIG drivers.** The performance of scMINER in signaling network inference and SIG driver identification was benchmarked using Perturb-seq (**a**–**e**) and CITE-seq (**f**–**h**) data as ground truth. **a** Network-level evaluations using the following metrics: number of true positive predictions as a function of top-ranked edges (left) and early precision ratio (middle) of top-ranked edges predicted by each method; ROC curves (right) of all predicted edges of scMINER (red) and random prediction (gray). **b** Heatmap showing the statistical significance of GSEA and AUROC of top 25 SIGs. **c** The regulon of IFNGR2 predicted by scMINER (n = 188). Red and blue edges indicate positive and negative correlations between IFNGR2 and its predicted downstream target genes. Edge color corresponds to the correlation strength measured by mutual information. Symbols of predicted target genes with |log$_2$FC | > 0.25 are shown (n = 25). **d** GSEA enrichment score curves of IFNGR2 for both positive (red)

and negative (blue) targets. **e** ROC curve of IFNGR2 regulon predicted by scMINER. The gray solid line represents a random prediction. AUROC: area under the receiver operating characteristic curve. **f** Bubble plot showing the expression, activity, and protein quantification by CITE-seq of cell type-specific markers in PBMCs. The cell type-specific markers were defined by differential analysis of protein quantification data, with cutoffs of P < = 0.01 and Fold change >= 2. The top 10 markers sorted by fold change were shown if over 10 markers were defined. The size of bubbles is proportional to the percentage of cells with quantification measurements, and the color of bubble is proportional to the scaled quantification measurements. **g, h** Violin plots (**g**) and scatter plots (**h**) showing the distribution of expression, activity by scMINER and protein quantification by CITE-seq of some known cell type-specific markers. Source data are provided as a Source Data file.

growth in scRNA-seq-based investigations. In addition, current version of scMINER is limited to estimating the activities of the drivers from curated lists of TF and SIG. While our current TF and SIG lists have been carefully curated and cover most currently known functional categories of driver candidates, they do lack consideration of novel non-TF or non-SIG drivers that are underexplored. This limitation might be solved by estimating the activity of genes using both their downstream and upstream first-order neighbors, since the networks inferred by scMINER encompass the entire transcriptome space. This solution would require further evaluation to be brought into practice. Furthermore, absence of ground-truth datasets for large-scale datasets limits the accuracy benchmark of scMINER, confining its performance benchmark to several smaller datasets in this study.

In summary, scMINER is a powerful tool for cell clustering, gene network inference, and hidden driver identification from scRNA-seq data. The increasing availability of single-cell transcriptomics datasets will provide ample opportunity for scMINER's application, including deciphering driver regulators of cell fate decisions, revealing the underlying mechanisms, and predicting therapeutic targets for cancer and other pathophysiological diseases.

## Methods

### scMINER Methods: Preface
scMINER is designed and optimized to provide a mutual information-based, end-to-end analysis of scRNA-seq data. This includes data intake, quality control and filtration, MI-based cell clustering, transcription factor (TF) and signaling (SIG) network inference, gene activity estimation, cell type annotation, differential expression and activity analysis, and data visualization and sharing. The scMINER framework is primarily developed using R for its advantages in statistical analysis and data visualization. It includes two additional components, MICA and SJARACNe[46], for cell clustering and network inference, respectively. MICA and SJARACNe are developed in Python to take its strengths in computational speed and memory management, as mutual information estimation of large-scale scRNA-seq data is typically compute-intensive.

### scMINER Methods: Key Concepts
**Mutual information.** Mutual information is a measure of the mutual dependence between two random variables. It quantifies the amount of information obtained about one variable through the other variable. In other words, it measures how much knowing the value of one variable reduces uncertainty about the value of the other. It's widely used in probability theory and information theory.

**Gene activity.** Gene activity estimation is a key feature of scMINER. Mathematically, the activity of one gene is a type of mean of the expressions of its targets. And biologically, it can be interpreted as a measure of how actively the driver functions, such as enzymes digesting their subtracts or kinases activating their downstream targets. Given gene expression profiles and networks, scMINER can

estimate the activities of pre-defined drivers, including not only transcription factors (TFs) but also signaling genes (SIGs).

**SparseEset.** The "Sparse ExpressionSet", or "SparseEset" for short, is a new class created by scMINER to efficiently store and access the sparse gene expression matrix of scRNA-seq data. Derived from the ExpressionSet class of Biobase[84] R package, the SparseEset object consists of the following data slots: 1) "assayData" for the sparse gene expression matrix, 2) "phenoData" for sample information, 3) "featureData" for feature information, and 4) annotation for dataset annotations. Similar to ExpressionSet class, the SparseEset object can be conveniently manipulated (e.g., subsetted, copied). scMINER also provides functions, *createSparseEset()*, *combineSparseEset()* and *updateSparseEset()*, to effortlessly create, combine, and update the SparseEset object, respectively.

### scMINER methods: reading input data
scMINER supports multiple input formats commonly used for scRNA-seq data. For data generated by the 10x Genomics Chromium platform, scMINER offers *readInput_10x.dir()* function to read directly from the standard CellRanger output directory, or *readInput_10x.h5()* function to read from the Hierarchical Data Format version 5 (HDF5) files using the hdf5r (v-1.3.11) R package. Additionally, scMINER can also read input data from H5AD files using the *readInput_h5ad()* function, which utilizes the anndata R package (v-0.7.5.6). scMINER also provides a function, *readInput_table()*, to read input data from text-table files, the most compatible text format for scRNA-seq data. These functions automatically convert the regular matrix into the sparse one using the Matrix (v-1.7-0) R package. They also allow modification of cell barcodes by adding prefixes or removing suffixes.

### scMINER Methods: Quality Control and Data Filtration
Quality control (QC) followed by data filtration is essential to remove the effects of low-quality cells from cell clustering and other analyses. In scMINER, we integrated four QC metrics that commonly used by the community[85]: nUMI (number of total UMIs in each cell), nFeature (number of expressed features/genes in each cell), pctMito (percentage of UMIs of mitochondrial genes defined by "MT-|mt-"), and pctSpikeIn (percentage of UMIs of spike-in RNAs defined by "ERCC-| Ercc-"). Additionally, we provided a metric for gene quality control: nCell (number of cells expressing each feature/gene). To facilitate quality assessment and cutoff determination, scMINER provides a function, *drawSparseEsetQC()*, to generate a HTML-format QC report summarizing key and detailed statistics of the five key metrics. The report also provides cutoffs automatically estimated by scMINER based on $Median \pm 3 \times MAD$ (maximum absolute deviance) and pseudo-filtration statistics with these cutoffs. scMINER offers another function, *filterSparseEset()*, to perform the data filtration efficiently and effortlessly. This function offers two modes for data filtration: "auto" and "manual". In the "auto" mode, scMINER uses the cutoffs automatically estimated by $Median \pm 3 \times MAD$, which are suitable for both

raw UMI counts and TPM values. And in the "manual" mode, users can manually specify the cutoffs for all five metrics.

## scMINER Methods: Cell-Cell Distance Estimation

The cell-cell distances are calculated by MICA, the cell clustering engine of scMINER, using MI:

$$I(C_i; C_j) = \sum_{y \in C_j} \sum_{x \in C_i} p_{(C_i, C_j)}(x, y) \log \frac{p_{(C_i, C_j)}(x, y)}{p_{C_i}(x) \cdot p_{C_j}(y)} \qquad (1)$$

where $C_i$ and $C_j$ are the $i$-th and $j$-th columns in gene expression matrix $M$. $p_{(C_i, C_j)}$ is the joint probability mass function of $C_i$ and $C_j$, and $p_{C_i}$ and $p_{C_j}$ are the marginal probability mass functions of $C_i$ and $C_j$ respectively.

Specifically, MICA first partitions the gene expression matrix $M$ into $M_{i,j}$ of a fixed number of cells. Then MI is estimated using the Eq. (1) in each $M_{i,j}$ in parallel. Since normalized gene expression values are continuous, we use a binning approach[86] for discretizing the expression values for joint and marginal probability calculations, where the bin size $b$ is defined as $\sqrt[3]{n}$ by default, with $n$ being the total number of genes. After that, MICA calculates the joint entropy of cells $C_i$ and $C_j$, $H(C_i, C_j)$, using the joint probability mass function $p_{(C_i, C_j)}$, as shown in the Eq. (2).

$$H(C_i C_j) = - \sum_{y \in C_j} \sum_{x \in C_i} p(C_i, C_j)(x, y) log_2 [p(C_i, C_j)(x, y)] \qquad (2)$$

After then, the MI matrices are merged and further normalized with the Eq. (3) as shown below:

$$D(C_i, C_j) = 1 - \frac{I(C_i; C_j)}{H(C_i, C_j)} \qquad (3)$$

where $H(C_i, C_j)$ is the joint entropy of cells $C_i$ and $C_j$, and $D(C_i, C_j)$ is the normalized cell-cell distance used in cell clustering analysis. scMINER provides a function, *generateMICAinput()*, to prepare the standard input for cell-cell distance estimation from the SparseEset object.

## scMINER Methods: dimensionality reduction and clustering

To balance the accuracy and efficiency, MICA encompasses two modes for cell clustering from a cell-cell distance matrix. The first mode, MICA-MDS, uses multidimensional scaling (MDS) to reduce dimensions, followed by k-means clustering. The number of dimensions is set to a fixed number of 19 by default as indicated in the benchmark analysis (Fig. 2e, f and Supplementary Fig. 3a). To reduce randomness, k-means clustering is run $n$ times ($n = 10$ by default) and the results of the multiple runs are aggregated using the consensus clustering method[87]. This mode is conducted by default for small datasets with 5,000 cells or less. The second mode, MICA-GE, is developed based on the graph embedding approach and is hence more scalable. In this mode, the mutual information matrix is represented as a $k$-nearest neighbor graph $G_1$ ($k = 80$ by default) using the HNSW algorithm (https://github.com/nmslib/hnswlib), followed by the node2vec embedding into a $d$-dimensional space ($d = 20$ by default). Then, an exact $k$-nearest neighbor graph $G_2$ ($k = 20$ by default) is constructed from the $d$-dimensional space using the ball tree algorithm[88]. The Louvain algorithm[89] is used as the default clustering method. This mode is recommended for datasets with 5,000 cells or more.

## scMINER Methods: UMAP and t-SNE Embedding

MICA supports both uniform manifold approximation and projection (UMAP) and t-distributed stochastic neighbor embedding (t-SNE) for single-cell embeddings. UMAP analysis is performed with the "umap" library, while t-SNE is conducted using the "sklearn.manifold" library. The scMINER R package offers a function, *addMICAoutput()*, to integrate MICA output into the SparseEset object, and the embedding results can be visualized using the *MICAplot()* function.

## scMINER methods: cell type annotation

scMINER supports both supervised and unsupervised strategies for cell type annotation. Supervised methods use known markers of potential cell types curated from existing studies, while unsupervised methods are based on differentially expressed genes. For supervised annotation, scMINER established a method that can estimate a signature score for each candidate cell type across all predicted clusters and integrated it in the *draw_bubbleplot()* function. Additionally, scMINER provides functions to visualize individual marker gene distribution, including *feature_vlnplot()*, *feature_boxplot()*, *feature_scatterplot()*, *feature_bubbleplot()* and *feature_heatmap()*. For unsupervised annotation, scMINER offers a function, *getDE()*, to identify cluster-specific differentially expressed genes.

## scMINER methods: network inference

The scMINER framework embeds a network rewiring engine, SJARACNe[46], for network inference. This process is composed of four key steps. First, the gene-gene dependency is measured using the adaptive partitioning method. Adaptive partitioning iteratively divides the two-dimensional space between each pair of genes into four quadrants, rather than binning the space into equally-sized bins, until the quadrant converges based on a $t$-test of significance. MI is calculated considering all quadrants. Then, the driver-target connections that are determined as non-statistically significant according to a threshold ($-pb = 1e^{-7}$ by default) or indirect interactions by applying a Data Processing Inequality (DPI) tolerance filter[90]. Next, a consensus network is built using a bootstrapping method. SJARACNe runs $n$ bootstraps ($-n = 100$ by default) and constructs $n$ MI networks from the first two steps. The consensus network is reconstructed by estimating the statistical significance of the frequency with which a specific edge is detected across all bootstrap runs, based on a Poisson distribution. Only statistically significant pairs are retained ($-pc = 1e^{-2}$ by default). Lastly, the consensus network is enhanced by adding the annotations of nodes and extra statistics such as Spearman and Pearson correlation coefficients, regression coefficients and associated P-values. To facilitate the network inference, scMINER provides the *generateSJARACNeInput()* function to prepare the standard input files for SJARACNe. This function supports downsampling and MetaCell strategy to better handle big datasets. scMINER provides another function, *drawNetworkQC()*, to comprehensively assess the quality of networks inferred by SJARACNe.

## scMINER methods: gene activity estimation

The gene activity estimated by scMINER is the mean of the gene expression values of all predicted targets. scMINER provides four types of means: 1) "mean", the arithmetic mean (default); 2) "weightedmean", the mean weighted by mutual information and signed by Spearman correlation coefficient; 3) "absmean", the mean of the absolute values of the expression measurements of all predicted targets; 4) "maxmean": the maximum value of the means calculated from positive or negative driver-target edges. scMINER provides two functions, *getActivity_individual()* and *getActivity_inBatch()*, to estimate gene activity for one or several cell populations, respectively. Both functions perform a column-wise Z-normalization to ensure each cell is on a similar expression level, followed by averaging the expression values of all predicted target genes with one of the four mean types mentioned above.

## scMINER methods: differential expression and activity analysis

scMINER provides two functions, *getDE()* and *getDA()*, for differential expression and activity analysis, respectively. Both functions embed three methods: "limma" from limma (v-3.60.4) R package, "wilcoxon" indicating the Wilcoxon Rank Sum test using stats (v-4.4.1) R package

and "t.test" indicating the Student's *t*-test using stats (v-4.4.1) R package. The *getDE()* function uses "limma" as the default method, while *getDA()* sets "t.test" as the default. In additional to *p* values, both functions return additional differential analysis measurements, including $\log_2$ fold change, FDR values calculated by *p.adjust()* function, Z scores transformed from *p* values using *combinePvaluVector()* function and signed by the $\log_2$ fold change values, and the percentage of cells without missing values on expression or activity.

### scMINER Methods: scMINER Portal

The scMINER Portal (https://scminer.stjude.org) is a web-based centralized data platform designed for visualizing, exploring, and sharing single-cell genomics studies of an extensive range of contexts. It is intended for researchers, data analysts, and anyone interested in single-cell data exploration and analysis. The scMINER Portal uses the Neo4j database system to manage the data of each study. In each Neo4j database, cells and genes are mapped to nodes, and the expression, activity, gene networks, and other measurements are designated as edge attributes. The back-end application of scMINER Portal is developed using the Java platform, specifically the Spring Framework, which is well-suited for high-performance enterprise-level application development. The front-end framework of scMINER Portal is built by Vue.js, with charts created using D3.js, Plotly.js, and CanvasXpress. Powered by these features, the scMINER Portal enables users to visualize and explore the datasets with one million cells or more efficiently and interactively.

### Benchmarking analysis of cell clustering: preface

The performance of scMINER in cell clustering was comprehensively benchmarked using three measures: accuracy, robustness, and efficiency. To accomplish this, we first curated 13 datasets generated by different protocols, varying in dimensions and the number of cell types (Supplementary Table 1). These datasets include 10 with ground-truth labels for accuracy benchmarking and 3 with at least 75,000 cells for efficiency evaluation. For comparative purposes, we selected 5 commonly-used methods of cell clustering that represent three common single-cell clustering methodologies[18]: Seurat and Scanpy (nearest-neighbor graphs), SC3s (consensus *k*-means), and scVI and scDeepCluster (deep-learning models). Additionally, we introduced various accuracy metrics that compare the predicted clusters to the true labels or assess the intrinsic properties of the clusters, including ARI (Adjusted Rand Index), AMI (Adjusted Mutual Information), NMI (Normalized Mutual Information), ASW (Average Silhouette Width), Accuracy Score defined as the weighted mean of the fraction of the major predicted labels in each true cell cluster, Purity Score defined as the weighted mean of the fraction of major true labels in each predicted cell cluster, and AvgBIO which is the average of ARI, NMI and ASW. For efficiency benchmarking, we measured runtime and peak memory usage metrics on a high-performance computing system with a 40-core Intel CPU across all 13 datasets.

The downloaded gene expression matrices of all 13 datasets are already filtered except the one of PBMC14K dataset. We manually removed the low-quality cells and genes of this dataset using the *filterSparseEset()* function with the default auto mode cutoffs (https://jyyulab.github.io/scMINER/bookdown/data-filtration.html#filter-the-sparse-eset-object).

### Benchmarking Analysis of Cell Clustering: scMINER

Subsequently, the raw UMI counts were normalized to CPM (Counts Per Million) and log2-transformed using the *normalizeSparseEset()* function. The cell clustering analysis was performed by MICA (v-1.0.0) following the standard tutorials (https://jyyulab.github.io/scMINER/bookdown/mi-based-clustering-analysis.html). The raw UMI counts were normalized to CPM (Counts Per Million) and log2-tranformed using the *normalizeSparseEset()* function. For the three datasets that

use TPM (Transcripts Per Million) as the gene expression measurement (Pollen, Chung and Usoskin), log2 transformation was performed directly without additional normalization. Then, the standard input file of each dataset for MICA was prepared using the *generateMICAinput()* function with default parameters. For datasets with 5,000 cells or fewer, the MICA-MDS mode was employed. All parameters were set to their default values, except for the *k* value of the *k*-means clustering analysis, which was set to match the number of cell populations according to the ground-truth labels. For datasets with more than 5,000 cells, the MICA-GE mode was used by default. The resolution for the Louvain method was swept from 0.1 to 4, with a step size of 0.1, to ensure that the predicted number of clusters matched the ground-truth labels. The result with *k* cluster was selected for subsequent analysis. If multiple resolutions returned the right number of clusters, the one of highest ARI was selected.

### Benchmarking analysis of cell clustering: seurat

Running of Seurat (version 5.0.1) for cell clustering followed the recommended tutorial (https://satijalab.org/seurat/articles/pbmc3k_tutorial). Specifically, the raw UMI counts were normalized to CP10K and log-transformed to log1p using the *NormalizeData()* function. For the three datasets using TPM (Transcripts Per Million) as the gene expression measurement (Pollen, Chung and Usoskin), log transformation was performed directly without additional normalization. Then, the top 2,000 HVGs were identified using the "vst" method embedded in the *FindVariableFeatures()* function, and used for PCA dimensionality reduction by *RunPCA()* function with default parameters (*npcs* = 50). The clustering analysis was conducted by Louvain method using the *FindClusters()* function. The resolution was swept from [0.1, 4] with a step size of 0.1 on the nearest-neighbor graph to ensure the predicted number of clusters matched the ground-truth labels. The result with *k* cluster was selected for subsequent analysis. If multiple resolutions returned the right number of clusters, the one of highest ARI was selected.

### Benchmarking analysis of cell clustering: scanpy

Running of Scanpy (version 1.10.2) for cell clustering followed the standard tutorial (https://scanpy.readthedocs.io/en/stable/tutorials/basics/clustering.html). The UMI count matrices were normalized to the median of total counts for cells using the *sc.pp.normalize_total()* function, followed by the log transformation conducted by the *sc.pp.log1p()* functioned. The top 2,000 HVGs were identified using the *sc.pp.highly_variable_genes()* function with default parameters. After the data scaling by *sc.pp.scale()*, PCA dimensionality reduction was conducted using the *sc.tl.pc()* function with $n_{comps}$ = 50. Leiden resolution was swept from [0.1, 4] with a step size of 0.1 in the clustering analysis performed by the *sc.tl.leiden()* function, to ensure the predicted number of clusters matched the ground-truth labels. The result with *k* cluster was selected for subsequent analysis. If multiple resolutions returned the right number of clusters, the one of highest ARI was selected.

### Benchmarking analysis of cell clustering: SC3s

Running of SC3s (version 0.1.1) for cell clustering followed its usage guidance (https://github.com/hemberg-lab/sc3s). The data preprocessing, normalization and dimensionality reduction were performed in the same way as Scanpy. The clustering analysis was conducted using the consensus *k*-means approach, with the *k* set to the number of ground- truth cell populations.

### Benchmarking analysis of cell clustering: scVI

Running of scVI (version 1.0.3) for cell clustering followed the guidelines from this tutorial (https://docs.scvi-tools.org/en/stable/tutorials/notebooks/quick_start/api_overview.html). The same methods and

parameters as Scanpy were applied for data preprocessing and normalization. scVI model was created using the *scvi.model.SCVI()* function and trained by the *model.train()* function with default parameters (*n_hidden:128, n_latent:10, n_layers:1*). The latent representation was used for building nearest-neighbor graph and identifying clusters via Leiden algorithm. Leiden resolution was swept from [0.1, 4] with a step size of 0.1 on the nearest-neighbor graph to ensure the predicted number of clusters matched the ground-truth labels. The result with $k$ cluster was selected for subsequent analysis. If multiple resolutions returned the right number of clusters, the one of highest ARI was selected.

### Benchmarking analysis of cell clustering: scDeepCluster

Running of scDeepCluster (version 1.0, pytorch version) (https://github.com/ttgump/scDeepCluster_pytorch) for cell clustering followed the recommended tutorial (https://github.com/ttgump/scDeepCluster_pytorch/blob/main/tutorial_10X_PBMC.ipynb). Similar to the scVI, the data preprocessing and normalization were performed using the same methods as Scanpy. scDeepCluster model was trained with default parameters and the latent representation was used for building nearest-neighbor graph. The clusters were identified by the consensus $k$-means algorithm, with the $k$ set to the number of ground-truth cell populations.

### Benchmarking analysis of network inference: preface

To comprehensively benchmark scMINER's performance in network inference, we curated four independent datasets, including two bulk ATAC-seq datasets and two CROP-seq dataset (Supplementary Table 2), and inferred the ground-truth edges of networks from them. For bulk ATAC-seq data, the ground-truth edges were defined by the TF-target pairs estimated by footprinting analysis. And for CROP-seq data, the ground-truth edges were established between the genes targeted by gRNA and differentially expression genes upon gRNA targeting. Next, we selected the three best-performing GRN inference tools in this benchmark study[17], GENIE3, GRNBoost2 and PIDC, and compared scMINER with them in retrieving the ground-truth edges inferred from the bulk ATAC-seq and CROP-seq datasets. Additionally, we employed multiple widely-used and well-accepted network evaluation metrics to make the benchmarking more comprehensive and robust.

### Benchmarking Analysis of Network Inference: Bulk ATAC-seq Data Analysis

The two bulk ATAC-seq datasets used in the network inference benchmarking analysis were downloaded from GSE123236[57] and GSE112731[58]. The GSE123236 dataset contains three lineages of CD8⁺ exhausted T cells: progenitor (Tpex), intermediate (Teff-like) and terminal (Tex). The GSE112731 dataset contains Foxp3⁺CD4⁺ regulatory T cells specific to spleen and visceral adipose tissue (VAT).

Both datasets were analyzed as described previously[91]. In brief, we first trimmed the raw reads with Trimmomatic (v-0.36, with parameters: *LEADING : 10TRAILING : 10TRAILING : 10SLIDINGWINDOW : 4: 18 MINLEN : 25*) and then aligned them to the reference genome mm10 (GRCm38) downloaded from GENCODE (release: M10) using BWA (v-0.7.16, with default parameters). Duplicated reads were marked by Picard (v-2.9.4, with default parameters) and removed by samtools (v-1.9, with parameters: *- q 1 - F* 1804). After the adjustment of Tn5 shift (reads were offset by +4 bp for the sense strand and −5 bp for the antisense strand), we selected the nucleosome-free reads (fragment size ≤146*bp*) and generated the 'bigwig' files by using the centre 80 bp of each fragment. Then, we performed the peak calling using MACS2 (v-2.1.1.20160309, with parameters: *−extsize200 − nomodel*), and generated consensus peaks by retaining peaks presenting in at least 50% of replicates. Next, we used FIMO from MEME suite[92] (v-4.11.3, with parameters: *−thresh0.0001 − motif − pseudo0.0001*) to scan motif

(TRANSFAC database release 2019, Vertebrata only) matches and assign the peaks to their nearest genes. Finally, we conducted footprinting analysis to identify the motif predicted binding sites (MPBS) of transcription factors with the RGT HINT[93] (v-2.7, with default parameters). For each MBPS, the ground-truth edges were established between the TFs of the matched motifs and the genes to which the MBPS was assigned.

### Benchmarking Analysis of Network Inference: CROP-seq Data Analysis

The two CROP-seq datasets used in the network inference benchmarking analysis were downloaded from GSE221321[59] and GSE218988[60]. The GSE221321 dataset was generated using a Perturbseq protocol, which transduced sgRNA libraries targeting 598 genes as well as non-targeting controls in THP-1 cells. The GSE218988 dataset was generated by an approach developed for both CRISPR interference (CRISPRi) and activation (CRISPRa) screens, which transduced a library containing 2,099 gRNAs in human CD8⁺ T cells.

For the GSE221321 dataset, we first evaluated the knockout (KO) or knockdown (KD) efficiency of all four samples involved and selected the conventional KD screen (GSM6858449) for subsequent analysis. Standard quality control cutoffs, calculated by median ± 3*MAD, were applied to library size, number of features, and mitochondrial gene content to remove the low-quality cells, as suggested in scater package[94] (v-1.32.1). Genes expressed in fewer than two cells were filtered out. Then, we performed differential expression analysis between cells with a single perturbation (only one gene targeted) and non-target (NT) cells using *findMarkers()* function from the scran package[95] (v-1.32.0). Only perturbated genes identified in 20 or more cells and showed significant KO/KD effect (log$_2$ fold change < −0.1) were retained. For each perturbed gene, targets were defined as genes with |log$_2$ fold change| ≥ 0.25 compared to NT cells.

For the GSE218988 dataset, the same methods and cutoffs were applied for data filtration, differential expression analysis, and perturbated gene filtration. Targets of each perturbated gene were defined based on the following criteria: log$_2$ fold change < −0.1 for CRISPRi and >0.1 for CRISPRa in the comparison of perturbed and NT cells. The two CROP-seq datasets used in the network inference benchmarking analysis were downloaded from GSE221321[59] and GSE218988[60]. The GSE221321 dataset was generated using a Perturbseq protocol, which transduced sgRNA libraries targeting 598 genes as well as non-targeting controls in THP-1 cells. The GSE218988 dataset was generated by an approach developed for both CRISPR interference (CRISPRi) and activation (CRISPRa) screens, which transduced a library containing 2099 gRNAs in human CD8⁺ T cells.

For the GSE221321 dataset, we first evaluated the knockout (KO) or knockdown (KD) efficiency of all four samples involved and selected the conventional KD screen (GSM6858449) for subsequent analysis. Standard quality control cutoffs, calculated by median ± 3*MAD, were applied to library size, number of features, and mitochondrial gene content to remove the low-quality cells, as suggested in scater package[94] (v-1.32.1). Genes expressed in fewer than two cells were filtered out. Then, we performed differential expression analysis between cells with a single perturbation (only one gene targeted) and non-target (NT) cells using *findMarkers()* function from the scran package[95] (v-1.32.0). Only perturbated genes identified in 20 or more cells and showed significant KO/KD effect (log$_2$ fold change < −0.1) were retained. For each perturbed gene, targets were defined as genes with |log$_2$ fold change| ≥ 0.25 compared to NT cells.

For the GSE218988 dataset, the same methods and cutoffs were applied for data filtration, differential expression analysis, and perturbated gene filtration. Targets of each perturbated gene were defined based on the following criteria: log$_2$ fold change < −0.1 for CRISPRi and >0.1 for CRISPRa in the comparison of perturbed and NT cells.

## Benchmarking analysis of network inference: scMINER

For the two bulk ATAC-seq datasets, matching scRNA-seq data of the three lineages of CD8[+] exhausted T cells and two tissue-specific Tregs were downloaded from GSE122712 and GSE130879, respectively. The network inference followed the standard scMINER pipeline (https://jyyulab.github.io/scMINER/bookdown/network-inference.html). In brief, we first preprocessed the raw count matrix of each dataset, including quality control, data filtration with default auto mode cutoffs and normalization. For the two CRISPR-perturbed datasets, QCed log2p1 CPM transformed expression matrices of NT cells were used for network inference (for in the case of TF and SIG network evaluation). We generated pseudobulk aggregates using SuperCell[47] (v-1.0, with parameters: gamma=20, n.pc=10). The standard input files for SJARACNe were generated using the *generateSJARACNeInput()* function in scMINER and SJARACNe was run in LSF mode with default parameters (-n 100, -pc 0.01) to infer both TF and SIG networks. The ground-truth network for each dataset was determined through differential expression (DE) analysis of these NT cells versus cells in which each TF was perturbed by different CRISPR technologies (e.g., knockout by CRISPR, activated by CRISPRa or inhibited by CRISPRi).

## Benchmarking analysis of network inference: GENIE3 and GRNBoost2

For the purpose of consistency, the same gene expression matrices used by scMINER were used as inputs for both GENIE3 and GRNBoost2 analysis. The list of hug genes, another input for GENIE3 and GRNBoost2, was prepared by extracting the transcription factors embedded in scMINER using the *getDriverList()* function. With these input files, we imported the arboreto.algo module from pySCENIC[25] package (v-0.12.1) and inferred the networks using the embedded *genie3()* and *grnboost2()* functions with default parameters.

## Benchmarking analysis of network inference: PIDC

Similar to GENIE3 and GRNBoost2, the same gene expression matrices preprocessed by scMINER and the hub genes extracted from scMINER embedded TFs were used for PIDC analysis. To make computation feasible, we trimmed the gene expression matrices of all datasets by selecting TF genes from scMINER database and top 7000 highly variable genes (HVGs) predicted by the variance stabilizing transformation (vst) method implemented in Seurat[19] (v-5.1.0). Network inference was conducted using the *PIDCNetworkInference()* function from NetworkInference PIDC Julia package (v-0.1.1, https://github.com/Tchanders/NetworkInference.jl) with default parameters.

## Benchmarking analysis of network inference: network evaluation metrics

To comprehensively benchmark the algorithms, we used the following metrics[17]: 1) Early precision (EP) and early precision ratio (EPR). We first ranked the edges by the scores (mutual information for scMINER and weights for others) for each method and then selected the top-$k$ edges for subsequent computation. EP was defined as the fraction of ground-truth edges identified by either bulk ATAC-seq or CROP-seq data in the top-$k$ edges. EPR was defined as the ratio of early precision value to the early precision for a random network. Both metrics were calculated using custom R scripts; 2) ROC and AUROC. Receiver operating characteristic curves were generated, and areas under these curves were calculated using the pROC[96] (v-1.18.5) R package with default parameters; 3) PRC and AUPRC. Precision-recall curves and areas under these curves were calculated using the PRROC[97] (v-1.3.1) R package with default parameters; 4) Gene set enrichment analysis (GSEA). As genes rank metric, we used log2 fold change values from the DE analysis of CROP-seq data (perturbed cells vs. NT. cells), and $p$ values were computed based on the Kolmogorov-Smirnov test using the *ks.test()* function from the stats R package (v-4.4.0). The $p$ values of positive and negative targets predicted by scMINER were merged using Fisher's

method implemented in the *sumlog()* function from the metap package (v-1.11). The random networks used in 1), 2), and 3) were generated with a uniform distribution of regulon size and random assignment of targets for each source gene. For global network evaluation based on ATAC-Seq data in the analysis of true positive edges and EPR values, the number of top-ranked edges to display was computed as a sum of a minimal number of targets per gene across evaluated networks (Tpex: 54,800, Teff-like: 60,182, Tex: 53,433, Tregs Spleen: 75,245, Tregs VAT: 80,193). ROC curves were computed for top edges in the global network (Fig. 4a, Supplementary Fig. 8a) and for all edges reported by each method (Supplementary Fig. 9). PRC curves for global networks were computed for top-ranked edges. AUROC and AUPRC values for each TF were computed for top targets for each source (Fig. 4a, Supplementary Fig. 8a) and for all genes reported by each method (Supplementary Fig. 9).

For activity evaluation of T-cell exhaustion drivers based on different TF networks, to ensure that networks are comparable, we considered top $n_i$ targets for each TF, where $n_i$ is the smallest number of targets for this TF among all evaluated networks. Activity was computed with a standard approach implemented in scMINER (as a mean value of Z-normalized expression values of TF targets in network). Markers were assigned to a cell type using a comparison of the activity of this marker among all pairs of cell types in the dataset. In each pair of compared cell types, the significance of the activity was estimated based on a one-tailed t-test with alternative hypotheses that activity is larger in one group of cells than in the other. The difference was considered significant if the P-value was less than 0.0001. A marker was assigned to a cell type if P-values were significant compared to all other cell types. To evaluate the accuracy of activity prediction for known markers, we computed true positive rate (TPR) of assigning markers to cell type in comparison to true label of each marker. All plots in this section were prepared with ggplot2 (v-3.5.1).

## Benchmarking analysis of activity estimation: preface

To comprehensively evaluate scMINER's performance in activity estimation, we performed three independent analyses. First, we evaluated how well the activities estimated by scMINER retrieved known drivers or markers in three datasets of different contexts: PBMC14K, CD8[+] exhausted T cells, and tissue-specific Tregs. To make the benchmarking more informative, we introduced SCENIC, another popular algorithm for estimating TF activities, for comparison. Second, we conducted the cell clustering using the activities estimated by scMINER and evaluated the accuracy by calculating the ARI using the aricode R package (v-1.0.3). We also benchmarked scMINER against SCENIC in this analysis. Third, we compared the activities estimated by scMINER with cell surface protein abundance using two CITE-seq datasets generated by different protocols. Since SCENIC only estimates TF activities, it was excluded from this analysis.

## Benchmarking analysis of activity estimation: scMINER

The first dataset, SRP073767[56], comprised seven subtypes of PBMC: monocytes, B cells, natural killer cells (NK), CD8+ naive T cells (CD8TN), CD4+ naive T cells (CD4TN), CD4+ regulatory T cells (CD4Treg), and CD4+ central memory T cells (CD4TCM). The second dataset, GSE122712[57], consisted of three subtypes of CD8+ exhausted T cells (TEX): progenitor exhausted T cells (Tpex), effector-like T cells (Teff-like), and terminally exhausted T cells (Tex). The third dataset, GSE130879[98], comprised four types of tissue-specific regulatory T cells (Tregs) from spleen, lung, skin, and visceral adipose tissue (VAT). For each of the three datasets, we estimated the activities of predefined TF and SIG drivers using the *getActivity_inBatch()* function with the default "mean" method. This function first performed a column-wise Z-normalization to ensure each cell was on a similar expression level. Then, for each driver, it calculated the mean value of gene expression measurements of all its predicted targets indicated in the cell type-specific

networks. Next, it concatenated the cell type-specific activity matrices and imputed missing values with the minimum non-NA value of the merged activity matrix.

### Benchmarking analysis of activity estimation: SCENIC

We performed activity estimation using the pySCENIC[25] package (v-0.12.1), following the standard workflow (https://pyscenic.readthedocs.io/en/latest/tutorial.html). First, we ran GRNBoost2 on the gene expression matrices preprocessed and used by scMINER to reconstruct the co-expression network with default parameters. Then, we used cisTarget for regulon identification. The following databases were used: hg19-RefSeq_r45-mc9nr motif collection for PBMC14K dataset (SRP073767), and mm10-RefSeq_r80-mc_v10 motif collection for CD8+ exhausted T cell dataset (GSE122712) and tissue-specific Tregs dataset (GSE130879). Next, we scored the regulon activity with the AUCell and generated the AUC matrix for subsequent analysis.

### Benchmarking analysis of activity estimation: activity-based cell clustering

As an accuracy metrics of activity estimation, activity-based cell clustering was performed on the three datasets used in benchmarks of network inference and hidden driver identification: PBMC14K, CD8+ exhausted T cells, and tissue-specific Tregs. For each dataset, we first generated the activity matrices using scMINER and SCENC separately, following their standard guidelines as mentioned above. For scMINER, we applied the MetaCell strategy using the *generateSJARACNeInput()* function (super-Cell_gamma = 50) and generated a global network with SJARACNe (-pc 0.01). The activity matrix, containing both TFs and SIGs, was generated by the *getActivity_individual()* function with the default "mean" method, and used as the input for activity-based clustering analysis. For SCENIC, the AUC matrix generated using AUCell was used as the input. Next, we conducted dimensionality reduction using the PCA method via *sklearn.decomposition.PCA()* function, and used the first 50 components in the Louvain clustering analysis. To ensure that the predicted number of clusters matched the ground-truth labels, we swept the Louvain resolution from 0.1 to 9 with a step size of 0.1. The resolution that generated the correct number of clusters was selected for ARI calculation using the aricode R package (v-1.0.3). If multiple resolutions returned the right number of clusters, the one of highest ARI was selected.

### Benchmarking analysis of activity estimation: CITE-seq data analysis

The two CITE-seq datasets were downloaded from Gene Expression Omnibus (GEO) under accession numbers: GSE213282[74] and GSE164378[75]. Both datasets were generated from human PBMCs using different protocols. The GSE213282 dataset was generated with the Chromium Next GEM Single Cell 5′ Protocol and contains 4,602 cells of 5 cell types. The GSE164378 dataset was generated with 10x Genomics 3′ v3 GEM kit and contains 161,764 cells of 8 cell types. Two ambiguous populations annotated as "other" and "other T" were manually excluded from subsequent analysis.

The analysis of both RNA and antibody-derived tags (ADT) data was performed using scMINER following the standard pipeline (https://jyyulab.github.io/scMINER). In brief, we first generated SparseEset objects for both RNA and ADT data separately using *createSparseEset()* function and filtered out the low-quality cells and genes with the default auto mode cutoffs. Then, we normalized and log2-transformed the filtered data using *normalizeSparseEset()* function with the default parameters. Next, the standard inputs for SJARACNe were generated by *generateSJARACNeInput()* function in a cell type-specific manner. The cell type annotations were collected from the source data. Default parameters were used for the GSE213282 dataset, while for the GSE164378 dataset, cell counts for each cell type were downsampled to 1,000 to improve efficiency. Network inference was performed by SJARACNe with default parameters: $-n100$, $-pc0.01$. After network

inference, the activity of predefined drivers was estimated using the *getActivity_inBatch()* function with the "mean" method. Markers were identified from ADT data using *getDE()* function with the "limma" method. DEGs with $p$ value < 1e-5 and fold change > 2 were defined as markers. The ADT, expression and activity data of these markers were visualized in a bubble plot generated by ggplot2 (v3.5.1). For the cell populations with over 10 markers defined, only the top 10 markers sorted by log2 fold change were shown.

### Reporting summary

Further information on research design is available in the Nature Portfolio Reporting Summary linked to this article.

## Data availability

The datasets for the cell clustering benchmark are summarized in Supplementary Table 1 and are available at Zenodo (https://doi.org/10.5281/zenodo.15040179). The datasets for benchmarking network inference and activity estimation are outlined in Supplementary Table 2. We additionally made the majority of these datasets available for interactive exploration at scMINER Portal (https://scminer.stjude.org). Source data are provided with this paper.

## Code availability

The source code of scMINER is freely available at GitHub (https://github.com/jyyulab/scMINER) and zenodo repository (https://doi.org/10.5281/zenodo.13224929[99]). An extensive tutorial and full documentation are available at https://jyyulab.github.io/scMINER. The codes for reproducing the majority of analyses in this paper can be found at https://github.com/jyyulab/scMINER_analysis/tree/main. The Neo4j-based interactive visualization portal, including documentation and software, is available at https://scminer.stjude.org.

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

## Acknowledgements

We thank the members of the Yu Lab for testing and improving scMINER, and Sarah August for scientific editing. This work was supported in part by National Institutes of Health grants RO1GM134382 (to J.Yu), U01CA264610 (to J.Yu), U01CA281868 (to H.C. and J.Yu), RO1CA274251 (to J.Yu), RF1AG068581 (to J.P.), and R35CA253188 (to H.C.), and by the American Lebanese Syrian Associated Charities. The content is solely the responsibility of the authors and does not necessarily represent the official views of the National Institutes of Health.

## Author contributions

Q.P., L.D., S.H., X.Yao, and J.Yu. conceived the project. Q.P., L.D., H.S., and J.Z. designed the computational methods, wrote software packages, and documentations. X.Yao, Y.D., S.H., and Q.P. carried out benchmarking analysis. L.Y. and Q.P. developed the data portal. H.S. and C.Q. contributed to software development and data analysis. X.D., C.B.,

J.P.V., A.K., M.R., and M.B. contributed to software development. Z.X., I.R., X.Yang, J.Yang, X.H., J.F., An.J., and Ar.J. assisted with data analysis, software testing, and portal development. K.K.Y. provided computational insights. J.P. and H.C. provided biological insights. Q.P., L.D., S.H., X.Yao, Y.D., and J.Yu wrote the manuscript. J.Yu supervised the project.

## Competing interests

L.D. is currently employed at Spatial Genomics Inc. All other authors declare no competing interests.

## Additional information

[1]Department of Computational Biology, St. Jude Children's Research Hospital, Memphis, TN 38105, USA. [2]Graduate School of Biomedical Sciences, St. Jude Children's Research Hospital, Memphis, TN 38105, USA. [3]Department of Immunology, St. Jude Children's Research Hospital, Memphis, TN 38105, USA. [4]Center for Molecular Medicine, Children's Hospital of Fudan University, Shanghai 201102, P.R. China. [5]Department of Information Services, St. Jude Children's Research Hospital, Memphis, TN 38105, USA. [6]Department of Physiology, University of Tennessee Health Science Center, Memphis, TN 38163, USA. [7]Precision Research Center for Refractory Diseases, Shanghai General Hospital, Shanghai Jiao Tong University School of Medicine, Shanghai 201620, China. [8]Department of Structural Biology and Developmental Neurobiology, St. Jude Children's Research Hospital, Memphis, TN 38105, USA. [9]These authors contributed equally: Qingfei Pan, Liang Ding, Siarhei Hladyshau, Xiangyu Yao. ✉e-mail: Jiyang.Yu@STJUDE.ORG

