## [Transparent Peer Review file · Nature Communications]

scMINER: a mutual information-based framework for clustering and hidden driver inference from single-cell transcriptomics data

Corresponding Author: Dr Jiyang Yu

Version 0:

Reviewer comments:

Reviewer #2

(Remarks to the Author)

The manuscript introduces scMINER, a framework that integrates existing tools for single-cell transcriptomic data analysis. While the paper does not propose fundamentally new methods, the integration of established approaches into a coherent pipeline is a valuable contribution, particularly in the context of unsupervised cell clustering, TF network inference, and SIG network analysis. By leveraging information theory to improve network inference, the approach offers an interesting alternative to traditional methods. Additionally, the authors provide comprehensive software documentation and tutorials, which significantly enhance the usability and accessibility of their approach for researchers. Nonetheless, despite the manuscript's overall clarity and the valuable contributions it makes, a few minor issues remain that warrant further attention.

Minor Comments:

1. Line 604: The equation for $H(C_i, C_j)$ is missing.
2. Line 608: This manuscript integrates several methods, and validating the optimality of each component in the combination is inherently difficult and nearly impossible. However, I am particularly concerned about the setting of the clustering modes. While the two modes, MICA-MDS and MICA-GE, offer distinct advantages for small and large datasets respectively, a direct comparison between the two strategies on a moderately sized dataset (e.g., ~5,000 cells) would be valuable to assess their performance trade-offs in terms of both accuracy and efficiency.
3. Line 613: "Extended Data Fig. 2a" may be "Extended Data Fig. 3a"
4. Line 730-734: Duplicated from Line 724-727.
5. Line 740: In benchmarking analysis of cell clustering, the authors swept the resolution of Louvain from 0.1 to 4 to ensure that the predicted number of clusters matched the ground-truth labels. A question arises: how can the resolution be determined in real data analysis when ground-truth labels are unavailable, or when only the number of clusters is known? One potential approach could be to perform a binary search to determine the resolution, which might be more efficient than the traditional grid search. Additionally, it would be valuable to know if the authors used parallel processing to accelerate the grid search process from 0.1 to 4. If so, how does this compare in terms of efficiency to binary search in both benchmarking and real data analyses?
6. Line 766, 784: While other methods use Louvain for benchmarking analysis of clustering, why did Scanpy opt for Leiden over Louvain? What advantages does Leiden offer in this context, and how does it compare to Louvain in terms of clustering performance?
7. Fig. 2e: The legend is missing.

8. Extended Data Fig. 4a: The y-axis should be adjusted to clearly display the performance of datasets excluding Klein.

(Remarks on code availability)

Reviewer #5

(Remarks to the Author)

I think the authors have addressed most of the comments from Reviewers 1 and 4, with the following points remaining:

1. Reviewer 1 has a comment "The choices of algorithms need to be explained more clearly and justified." The authors responses to this comment, in terms of the choice of clustering methods, is rather high-level: "The clustering engine, MICA, incorporates methods that were selected after thorough evaluation of available algorithms for cell clustering, particularly in dimensionality reduction." It would be more clear if the authors can elaborate and present the specific evaluation that was performed. The manuscript does not provide more details on this. Although two benchmarking papers were cited, it is not straightforward which top performing methods were adapted in scMINER. Specific information needs to be provided to justify the choice of methods.

2. In the response to Reviewer 4's comments, the authors stated that "Due to the size limitation of the R package, it's not feasible to include the input data, codes and documentation for all benchmarked datasets. Instead, we have used the PBMC14K dataset as a showcase and have integrated its input data and codes in the scMINER R package. " I suggest that the authors explore the possibility of depositing raw or preprocessed data in Zenodo, for datasets presented in the paper.

3. If submitting another revision, it would be helpful if the authors can highlight changes in the manuscript and in the responses refer to line numbers in the manuscript so that reviewers can quickly locate the changes corresponding to each response.

(Remarks on code availability)

I tested the interactive platform and basic functions are working.

Reviewer #6

(Remarks to the Author)

Pan et al., detailed the development of a very useful tool, scMINER, that can perform cell clustering, transcription factor and signaling protein network inference analyses, and identify hidden drivers from single cell transcriptomic data analyses that not only have a much better performance compared to other existing methods that have some of the similar functions, but also provide novel functions, such as signaling protein network analysis. The most challenging tests are the applications on T cell subsets that normally can only be distinguished by transcriptional factors, mostly lowly expressed. The authors thoroughly addressed the reviewers' critiques raised previously. I do not have any major concerns and only some minor points:

1. A few abbreviations need to be defined.

2. Grammatic errors need to be corrected.

3. The abstract and the last paragraph of the intro missed an opportunity to highlight innovation and extensive validation that this study provides.

(Remarks on code availability)

Point by point responses to reviewer's comments:

We sincerely appreciate the insightful and constructive comments from all reviewers. Below, we provide our detailed, point-by-point responses to their comments.

Reviewer #2 (Remarks to the Author)

The manuscript introduces scMINER, a framework that integrates existing tools for single-cell transcriptomic data analysis. While the paper does not propose fundamentally new methods, the integration of established approaches into a coherent pipeline is a valuable contribution, particularly in the context of unsupervised cell clustering, TF network inference, and SIG network analysis. By leveraging information theory to improve network inference, the approach offers an interesting alternative to traditional methods. Additionally, the authors provide comprehensive software documentation and tutorials, which significantly enhance the usability and accessibility of their approach for researchers. Nonetheless, despite the manuscript's overall clarity and the valuable contributions it makes, a few minor issues remain that warrant further attention.

Minor Comments:

1. Line 604: The equation for $H(C_i, C_j)$ is missing.

Response: We thank the reviewer for catching this. We have added the equation for $H(C_i, C_j)$ in the revised manuscript (**Line 602-605**).

2. Line 608: This manuscript integrates several methods, and validating the optimality of each component in the combination is inherently difficult and nearly impossible. However, I am particularly concerned about the setting of the clustering modes. While the two modes, MICA-MDS and MICA-GE, offer distinct advantages for small and large datasets respectively, a direct comparison between the two strategies on a moderately sized dataset (e.g., ~5,000 cells) would be valuable to assess their performance trade-offs in terms of both accuracy and efficiency.

Response: We appreciate the reviewer's insightful comments regarding this aspect of our study. As described in the manuscript (**Line 120-122**), we developed two operational modes to optimize clustering accuracy while ensuring scalability for large datasets. Our benchmarking results indicate that the MICA-MDS mode achieves superior clustering accuracy, whereas the MICA-GE mode excels in handling large datasets with efficiency.

Following the reviewer's suggestion, we conducted a comparative analysis of these two modes using a dataset of ~5,000 cells. This dataset was derived from the PBMC14K dataset by randomly selecting 750 cells for each of the seven cell types. Clustering analysis was performed under both modes using default parameters. The result, summarized in the table below, demonstrated that the MICA-MDS mode achieved an ARI of 0.833, slightly outperforming the MICA-GE mode with an ARI of 0.828. On the other hand, the MICA-GE mode exhibited significantly better computational efficiency

than MICA-MDS, with a runtime of 734 seconds compared to 4,332 seconds for the MICA-MDS mode.

Response Table 1. Comparison of MICA-MDS and MICA-GE modes on a medium-size dataset.

Dataset	Mode	ARI	Running time	Max Memory
PBMC cells (750 cells * 7 cell types = 5250 cells)	MICA-MDS	0.833	4332 sec.	1235 MB
	MICA-GE	0.828	734 sec.	3242 MB

We would also like to clarify that the 5,000-cell threshold for mode selection is an empirical guideline rather than a strict cutoff. The scMINER framework allows users to adjust this threshold based on their specific dataset characteristics and computational requirements.

3. Line 613: “Extended Data Fig. 2a” may be “Extended Data Fig. 3a”

Response: We thank the reviewer for catching this error. It has been corrected in the revised manuscript (**Line 617**).

4. Line 730-734: Duplicated from Line 724-727.

Response: We apologize for the confusion. Now the redundant text has been removed.

5. Line 740: In benchmarking analysis of cell clustering, the authors swept the resolution of Louvain from 0.1 to 4 to ensure that the predicted number of clusters matched the ground-truth labels. A question arises: how can the resolution be determined in real data analysis when ground-truth labels are unavailable, or when only the number of clusters is known? One potential approach could be to perform a binary search to determine the resolution, which might be more efficient than the traditional grid search. Additionally, it would be valuable to know if the authors used parallel processing to accelerate the grid search process from 0.1 to 4. If so, how does this compare in terms of efficiency to binary search in both benchmarking and real data analyses?

Response: We thank the reviewer for raising this fundamental question and for suggesting potential solutions. Determining the optimal resolution is indeed a critical and challenging task, as it requires balancing clustering granularity with biological interpretability. This balance usually cannot be achieved without subsequent analyses, such as marker gene expression and functional annotation. However, there are several approaches to address this challenge, and we have equipped scMINER with most of them, including but not limited to the following:

1) Iterative Resolution Testing

scMINER provides a user-friendly way to perform iterative resolution testing, simpler than what is offered by tools like Seurat and Scanpy. Users can define a range of resolutions by specifying the minimum, maximum and step-size. scMINER processes all resolutions in parallel to ensure computational efficiency

and generates clustering results for all test resolutions, making it easier to compare and evaluate the outcomes.

2) Clustering Evaluation Metrics

scMINER supports various metrics for clustering evaluation. For example, the Silhouette Score can be calculated for each resolution, which helps assess how well the clusters were defined.

3) Biological Interpretability Tools

scMINER embeds several functions to enhance biological interpretability, such as marker gene identification and visualization. These features are designed to help users better understand the biological relevance of the clusters. More details can be found here: <https://jyyulab.github.io/scMINER/bookdown/cell-type-annotation.html>.

For real-world datasets where ground-truth labels or the number of clusters are not available, we recommend users to leverage these functionalities to generate clustering results across multiple resolutions. By comparing the results for both granularity and biological interpretability, users can select the resolution that best balances these two aspects.

Regarding the binary search algorithm, while it is an efficient method for finding a target value in a sorted array, it may not be suitable in this context. This is because the relationship between clustering resolution and the number of clusters is not always linear, making binary search less applicable for this specific problem.

6. Line 766, 784: While other methods use Louvain for benchmarking analysis of clustering, why did Scanpy opt for Leiden over Louvain? What advantages does Leiden offer in this context, and how does it compare to Louvain in terms of clustering performance?

Response: We thank the reviewer for raising these questions. The Leiden algorithm was introduced as an improvement over the Louvain algorithm (PMID: 30914743). It addresses several known limitations of the Louvain algorithm, including its inability to guarantee well-connected communities, suboptimal modularity, and the resolution limit. Theoretically, the Leiden algorithm offers better clustering accuracy than the Louvain method and is recommended in the standard tutorial of Scanpy (<https://scanpy.readthedocs.io/en/stable/tutorials/basics/clustering.html>). At the reviewer's suggestion, we compared the clustering performance of these two methods using all 10 ground-truth datasets. For these analyses, all inputs and parameters were kept identical, except for the clustering method. As shown in **Response Figure 1** below, the Leiden algorithm outperformed the Louvain method in clustering accuracy for 7 out of 10 datasets (**left panel**), while the Louvain method demonstrated slightly better clustering efficiency than the Leiden algorithm across most dataset (**right panel**). These results align with findings on clustering accuracy and efficiency from other benchmarking studies (PMID: 30914743).

7. Fig. 2e: The legend is missing.

Response: We thank the reviewer for catching this. We have added the legend (Line 1298-1301) in our revised manuscript.

8. Extended Data Fig. 4a: The y-axis should be adjusted to clearly display the performance of datasets excluding Klein.

Response: We appreciate the reviewer's valuable suggestion. In response, we have updated the y-axis scale using the log₁₀-transformation method to enhance the clarity and interpretability of the data. The revised figure can now be found in **Supplementary Fig. 4a**. For the reviewer's convenience, we have also included the updated figure below (**Response Figure 2**).

Reviewer #5 (Remarks to the Author)

I think the authors have addressed most of the comments from Reviewers 1 and 4, with the following points remaining:

1. Reviewer 1 has a comment "The choices of algorithms need to be explained more clearly and justified." The authors responses to this comment, in terms of the choice of clustering methods, is rather high-level: "The clustering engine, MICA, incorporates methods that were selected after thorough evaluation of available algorithms for cell clustering, particularly in dimensionality reduction." It would be more clear if the authors can elaborate and present the specific evaluation that was performed. The manuscript does not provide more details on this. Although two benchmarking papers were cited, it is not straightforward which top performing methods were adapted in scMINER. Specific information needs to be provided to justify the choice of methods.

Response: We thank the reviewer for bring this to our attention and apologize for any confusion caused by the previous description, which may have been unclear or insufficient. Here, we provide more details about the algorithms incorporated in scMINER and the specific evaluations conducted for each.

- **Algorithm #1: MICA, the cell clustering engine of scMINER**

When developing MICA, we focused on two key aspects of cell clustering analysis: cell-cell distance estimation and dimensionality reduction. For cell-cell distance estimation, we evaluated four widely-used metrics: mutual information (MI), Pearson correlation coefficient, Spearman rank correlation coefficient and Euclidean distance. We performed cell clustering analysis using these four metrics on four gold-standard ground-truth datasets (Yan, Pollen, Kolod, and Buettner), while keeping the other steps and parameters the same. As shown in **Response Figure 3a,b** (same to **Fig. 2f** and **Supplementary Fig. 3a**), MI achieved the highest ARI. Similarly, we assessed four common dimensionality reduction methods: multidimensional scaling (MDS), PCA, Laplacian eigenmaps and a combination of PCA and Laplacian (PCA/Laplacian) using the same ground-truth datasets. As shown in **Response Figure 3c,d** (same to **Fig. 2g** and **Supplementary Fig. 3a**), MDS outperformed the other methods in clustering accuracy. Based on these specific evaluations, we choose to integrate MI-based cell-cell distance estimation method and MDS-based dimensionality reduction method into MICA.

- Algorithm #2: SJARACNe, the network inference engine of scMINER.** SJARACNe is re-engineered version of ARACNe, optimized to efficiently reconstruct gene networks from large datasets (PMID: 30388204). The accuracy of network inference by scMINER has been thoroughly benchmarked against three well-established tools-GENIE3, GRNBoost2 and PIDC-using datasets from different modalities, including ATAC-seq, Perturb-seq and CITE-seq. Due to space constrains, we have not included these results here, but they can be found in **Fig. 4**, **Fig. 6**, **Supplementary Fig. 8** and **Supplementary Fig. 9**.

We would also like to clarify that the top-performing methods discussed in the two benchmarking papers (PMID: 35135612; PMID: 31907445) are included for

benchmarking purposes only. scMINER does not incorporate any components of these methods.

2. In the response to Reviewer 4's comments, the authors stated that "Due to the size limitation of the R package, it's not feasible to include the input data, codes and documentation for all benchmarked datasets. Instead, we have used the PBMC14K dataset as a showcase and have integrated its input data and codes in the scMINER R package. " I suggest that the authors explore the possibility of depositing raw or preprocessed data in Zenodo, for datasets presented in the paper.

Response: We thank the reviewer for this instructive suggestion. Following this suggestion, we have uploaded the raw and preprocessed data of all datasets used in clustering benchmarks to the Zenodo (DOI: [10.5281/zenodo.15040179](https://doi.org/10.5281/zenodo.15040179)).

3. If submitting another revision, it would be helpful if the authors can highlight changes in the manuscript and in the responses refer to line numbers in the manuscript so that reviewers can quickly locate the changes corresponding to each response.

Response: We thank the reviewer for this thoughtful suggestion and sincerely apologize for any inconvenience caused the absence of highlighted changes in the previous revised manuscript. The decision not to highlight changes was made because over 60% of the text had been refined, which would have resulted in excessive and potentially distracting annotations. However, in line with the reviewer's suggestion, we have now specified the **line numbers** in our responses and highlighted all changes in a **red font** in the newly revised manuscript for easier review and clarity.

4. Remarks on code availability: I tested the interactive platform and basic functions are working.

Response: We thank the reviewer for testing and confirming the functions of our scMINER portal.

Reviewer #6 (Remarks to the Author)

Pan et al., detailed the development of a very useful tool, scMINER, that can perform cell clustering, transcription factor and signaling protein network inference analyses, and identify hidden drivers from single cell transcriptomic data analyses that not only have a much better performance compared to other existing methods that have some of the similar functions, but also provide novel functions, such as signaling protein network analysis. The most challenging tests are the applications on T cell subsets that normally can only be distinguished by transcriptional factors, mostly lowly expressed. The authors thoroughly addressed the reviewers' critiques raised previously. I do not have any major concerns and only some minor points:

1. A few abbreviations need to be defined.
2. Grammatic errors need to be corrected.

Response (to 1 and 2): We thank the reviewer for the reminders on abbreviations and grammatic errors. In response, we have thoroughly reviewed the manuscript to ensure that all abbreviations are clearly defined wherever necessary and that any grammatical errors have been carefully corrected. The relevant modifications have been made and can be found at the following locations: **Line 118-119, Line 174-175, Line 1316-1318, Line 1327-1330, Line 1344, Line 1367-1368, Line 1382**. For clarity and ease of review, these changes have been highlighted in red font in the revised manuscript.

3. The abstract and the last paragraph of the intro missed an opportunity to highlight innovation and extensive validation that this study provides.

Response: We sincerely thank the reviewer for this thoughtful suggestion and agree that emphasizing the innovation and extensive validations in the Abstract and Introduction would strengthen the manuscript. However, due to the space constrains, providing detailed descriptions in both sections might make them overly text-heavy. Instead, we have included a comprehensive discussion of these aspects in the first paragraph of the Discussion section (**Line 471-483**). We believe this placement effectively conveys the significance of our work while adhering to the formatting requirements.